# It Takes Two to Tango: A Review of Oncogenic Virus and Host Microbiome Associated Inflammation in Head and Neck Cancer

**DOI:** 10.3390/cancers14133120

**Published:** 2022-06-25

**Authors:** Mallory G. McKeon, Jean-Nicolas Gallant, Young J. Kim, Suman R. Das

**Affiliations:** 1Division of Infectious Diseases, Department of Medicine, Vanderbilt University Medical Center, 1161 21st Avenue South, Medical Center North, Suite A2200, Nashville, TN 37232, USA; mallory.g.mckeon@vanderbilt.edu; 2Department of Otolaryngology—Head and Neck Surgery, Vanderbilt University Medical Center, Nashville, TN 37232, USA; jn.gallant@vumc.org (J.-N.G.); y2.kim@vumc.org (Y.J.K.); 3Department of Pathology, Microbiology and Immunology, Vanderbilt University Medical Center, Nashville, TN 37232, USA

**Keywords:** pharynx, oncogenic virus, oropharyngeal cancer, nasopharyngeal cancer, human papillomavirus (HPV), Epstein–Barr virus (EBV), microbiome, infection, inflammation, carcinogenesis

## Abstract

**Simple Summary:**

Certain viruses, specifically, human papillomavirus (HPV) and Epstein–Barr virus (EBV), have been linked with the development of head and neck cancer. In this study, we review the mechanisms by which (these) viruses lead to cellular transformation and a chronic inflammatory state. Given that the head and neck host a rich microbiome (which itself is intrinsically linked to inflammation), we scrutinize the literature to highlight the interplay between viruses, cellular transformation, inflammation, and the local host microbiome in head and neck cancer.

**Abstract:**

While the two primary risk factors for head and neck squamous cell carcinoma (HNSCC) are alcohol and tobacco, viruses account for an important and significant upward trend in HNSCC incidence. Human papillomavirus (HPV) is the causative agent for a subset of oropharyngeal squamous cell carcinoma (OPSCC)—a cancer that is impacting a rapidly growing group of typically middle-aged non-smoking white males. While HPV is a ubiquitously present (with about 1% of the population having high-risk oral HPV infection at any one time), less than 1% of those infected with high-risk strains develop OPSCC—suggesting that additional cofactors or coinfections may be required. Epstein–Barr virus (EBV) is a similarly ubiquitous virus that is strongly linked to nasopharyngeal carcinoma (NPC). Both of these viruses cause cellular transformation and chronic inflammation. While dysbiosis of the human microbiome has been associated with similar chronic inflammation and the pathogenesis of mucosal diseases (including OPSCC and NPC), a significant knowledge gap remains in understanding the role of bacterial-viral interactions in the initiation, development, and progression of head and neck cancers. In this review, we utilize the known associations of HPV with OPSCC and EBV with NPC to investigate these interactions. We thoroughly review the literature and highlight how perturbations of the pharyngeal microbiome may impact host-microbiome-tumor-viral interactions—leading to tumor growth.

## 1. Introduction

The distinct microenvironments of the human upper aerodigestive tract host rich microbiomes that bridge the external environment and body. Perhaps the best studied of these upper aerodigestive tract ecosystems is the oral microbiome. Evidence shows that a healthy and diverse oral microbiome can protect against a wide array of foreign pathogens and host diseases [1,2]. The oral microbiome is comprised of a common core bacterial component, variation from which may be associated with behavioral, geographic, temporal, or health-related factors [1,2,3]. For example, long term use of tobacco products generates predictable shifts in the oral microbiome that have been linked to oral cavity cancer. These shifts include overgrowth of specific oral bacteria, namely *Porphyromonas**, Prevotella,* and *Fuscobacterium* species, that lead to the development of oral squamous cell carcinoma [4]. Chronic periodontal disease also causes similar shifts in the oral microbiome and leads to oral cavity cancer [5]. Though the oral microbiome and its link to oral cavity cancer has been well studied, less is known about the pharyngeal microbiome and its link to pharyngeal cancers [6].

Part of the complexity of the pharyngeal microbiome (with regards to cancer) is due to the fact that many viruses colonize pharyngeal mucosa [7]. The implications of chronic latent infection, subsequent chronic inflammatory response in these tissues, and malignant transformation is unclear (and the subject of this review). Two pertinent examples are the cancer-related viruses, human papillomavirus (HPV) and Epstein–Barr virus (EBV), which colonize the pharynx and are responsible for a growing proportion of head and neck cancers [8]. Seventy to ninety percent of all oropharyngeal squamous cell carcinomas (OPSCCs) are linked to HPV, especially the high-risk HPV16 genotype [9,10,11]. Likewise, nearly all patients with nasopharyngeal carcinoma (NPC) have been exposed to EBV, and EBV transcripts (or positive serology) are detected in about ninety percent of such patients (more frequently in the non-keratinizing, as opposed to keratinizing, subtype of NPC) [12,13,14]. It usually takes decades after being infected with HPV or EBV for cancer to develop [15]. It is estimated that approximately 15% of adults have a high-risk HPV oropharyngeal infection in their lifetime, with many more likely having some type of exposure given infection rates at other anatomic sites [11,16,17]. Similarly, 95% of adults are either infected or exposed to EBV during their lifetime [18]. Despite this high infection prevalence, only a small percentage progress to cancer. Thus, HPV or EBV appear to be necessary but insufficient causes of cancer, and other mediators, such as the host’s microbiome—or the immune response to both the viral infection and the microbiome—may be associated with carcinogenesis [19].

Chronic inflammation indeed plays a prominent role in carcinogenesis and tumor growth in the oral cavity [20]. This pattern has been reflected in other settings with specific associations with viral infection, such as hepatitis C and hepatocellular carcinoma [12,20]. The human immune system generally utilizes inflammatory cytotoxic T cells and interferons to recognize and respond to viral infection like HPV and EBV [21,22]. However, both HPV and EBV have developed mechanisms to evade detection and prevent destruction of infected cells (more on this below) [23,24]. Despite evasion of innate immune components, inflammatory response still may be initiated in the setting of HPV and EBV viral infection. However, it is unclear if the direct viral infection itself or the modification of the microbiome community mediates these changes in chronic inflammatory responses (Figure 1).

A healthy microbiome reduces inflammation and allows for normal mucosal function. Since an inflammatory response can promote transformation, and because pharyngeal (e.g., EBV, HPV) viral infections can evade immune detection via inflammation and tumor modulating genes expression, the question is whether this combination/milieu is synergistically oncogenic. In this review, we examine this question by evaluating the literature at the edge of the immune system, the microbiome, and oncogenic viruses in the upper aerodigestive tract. We specifically examine this relationship through the lens of HPV/OPSCC and EBV/NPC.

## 2. Methods

In order to identify studies linking the inflammation, the microbiome, EBV/HPV, and head and neck cancer, we conducted a systematic-style search in PubMed. The terms pharynx, oropharynx, nasopharynx (and related anatomy), viral infections, inflammation, EBV, HPV, and carcinogenesis were queried. The specific search query was: (“Pharynx” [Mesh:NoExp] OR “Oropharynx” [Mesh] OR “Nasopharynx” [Mesh] OR “Palate, Soft” [Mesh] OR “Eustachian Tube” [Mesh]) AND(“Viruses” [Mesh] OR “Virology” [Mesh] OR “Virome” [Mesh] OR “Infections” [Mesh] OR “Inflammation” [Mesh] OR “Epstein-Barr Virus Infections” [Mesh] OR “Papillomaviridae” [Mesh] “Epstein Barr” [tiab] OR “EBV infection” [tiab] OR “EBV infections” [tiab] OR papillomaviridae [tiab] OR “human papilloma virus” [tiab] OR “human papilloma viruses” [tiab] OR “human papillomavirus” [tiab] OR HPV [tiab]) AND (“Carcinogenesis” [Mesh] OR “Oropharyngeal Neoplasms” [Mesh] OR “Nasopharyngeal Neoplasms” [Mesh]). No date limits were imposed, only complete entries were considered, and the last search was performed on 21 October 2021.

Our search initially resulted in 284 records. Titles were then independently evaluated for relevance by authors JG and MGM (using a citation manager), resulting in 114 relevant abstracts. Whole manuscripts were further screened for relevance. Non-original articles, review articles, non-human studies, and abstract-only publications were then excluded. Additionally, studies published in languages other than English were excluded. Interrater conflicts were assessed and re-designated relevant or irrelevant by MGM, leaving 45 articles for review. Any pertinent topic areas that were uncovered following this thorough search were supplemented with specific searches in complementary databases, such as Google Scholar, PubMed, and Web of Science. Additional searches to specifically link the microbiome, HPV, and OPSCC yielded 6 pertinent articles; no articles were found linking the microbiome, EBV, and NPC.

## 3. Results

### 3.1. Human Papillomavirus and Oropharyngeal Carcinoma

Human papillomavirus (HPV) is a nonenveloped, double stranded, circular DNA virus that is primarily transmitted through direct contact with viral fomites [34,35,36]. The virus infects the proliferative basal layer of human epithelial cells and depends on epithelial replication and differentiation to complete its lifecycle (Figure 2) [16,24]. Early expression of viral proteins during the quiescent phase of infection promotes preferential replication of infected cells, facilitating lateral expansion along the basement membrane. HPV, including its numerous high-risk strains, is relatively common in humans [11]. Most infections are cleared within six months to a year; however, persistent infection with high-risk HPV is the one of the primary etiologic factors for cervical cancer, as well as other anogenital and head and neck cancers [10,16,37,38]. Specifically, approximately 70% of these cancers are caused by HPV16 and HPV18 genotypes [10,11,39,40]; these strains are detected significantly more often in biopsies of malignant tonsillar disease than in benign tonsillar pathology [41,42].

Oral cavity squamous cell carcinoma (OCSCC) and oropharyngeal squamous cell carcinoma (OPSCC) are two of the most common malignancies of the head and neck, with high morbidity and mortality in affected patients [51]. Though anatomically different, they share many risk factors. For example, tobacco and alcohol use independently exhibit dose dependent risk amplification for OCSCC and OPSCC [52]. However, the two diseases are distinct in that HPV is implicated in a majority of new OPSCC cases versus only 7–14% of OCSCC [10,11,51]. This difference is significant enough that it causes widely different presentation of disease (HPV positive OPSCC presents with smaller primary tumors and more advanced lymphatic spread) and affects its treatment (HPV positive OPSCC is often treated with chemoradiation alone due to high response rates) [53,54].

Over the past two decades, there has been a growing population of OPSCC linked to HPV infection in young white males who do not report alcohol or tobacco history [55]. The question thus arises as to whether increasing rates of OPSCC in this population are due to rising HPV infection rates or another etiologic factor. Investigation of archived tonsillar tissue has demonstrated a notable lack of high-risk HPV16 in samples from cohorts prior to the year 2000, suggesting a rise in infection rate associated with the observed rise in OPSCC development [56]. Hypotheses abound as to why the white male demographic is more significantly suffering from these shifts include changing sexual behaviors, decreasing tobacco use, and biological variation in viral clearance [57]. Studies evaluating the prevalence of active oral HPV infection have found that 1–5% of adults have HPV16 detectable in their saliva at any one time [58,59]. In contrast to active infection, estimates are that 90% of adults have been exposed to HPV16 and 70% have evidence of infection as demonstrated by the presence of HPV16 antibodies in their blood [11]. Despite this high infection prevalence, only a small percentage progress to cancer [11]. Thus, HPV is the necessary but insufficient cause of cancer, and other mediators like host’s chronic inflammation have been proposed to be associated with pathogenesis [19].

### 3.2. Cellullar Mechanisims of HPV-Mediated Oropharyngeal Carcinogenesis

Research characterizing the pathogenesis of HPV-mediated disease in the oropharynx is relatively new, as compared to other anatomic sites of HPV infection (i.e., the cervix). However, these other sites of disease have yielded much information about the pathogenesis of HPV-induced carcinogenesis. While dysplasia, as seen in premalignant lesions of the cervix, is not observed in the oropharynx, the same three primary viral proteins, E5, E6, and E7, have been demonstrated to be key players in HPV positive OPSCC (Figure 2) [24,36].

E5 is a viral protein produced early in the infectious lifecycle of HPV. E5 complexes with the epidermal growth factor receptor (EGFR) to stimulate growth and replication of infected cells [43]. E5 also blocks degradation of EGFR proteins, perhaps by manipulation of the cytoskeleton or decreasing acidity inside of lysosomes [40]. Amplification of other growth factors, namely fibroblast growth factor receptor 1, has been implicated in pathogenesis of tonsillar SCC, but this is not limited to HPV-positive cases [60]. Additionally, E5 down regulates CD-1d and MHC to avoid identification by natural killer cells and cytotoxic T cells [45]. Stimulation of EGFR also activates down steam products including prostaglandins, COX2, and PGE2, which promote angiogenesis through VEGF [61]. Although it has been demonstrated to prevent apoptosis in these cells, the E5 transcript is deleted relatively early in the infectious cycle, and therefore is not thought to contribute to carcinogenesis [62].

E6 is one of the main drivers of viral proliferation [40]. This protein downregulates p53 and BAK, both key proapoptotic proteins [63]. Inaction of these proteins leads to chromosome instability. Degradation of p53 is unique to high-risk strains of HPV, such as HPV16 [64]. Additionally, E6 has been shown to activate telomerase and stimulate molecular signaling, such as Wnt and other cancer hallmark pathways, in infected cells [65]. In HPV positive human oral keratinocytes and tonsillar epithelial cells, the FOXM1B oncogene and its upstream regulator, GRHL2, are directly induced by E6 [66]. Both FOXM1B and GRHL2 are absent in normal oral keratinocytes and tonsillar epithelium, suggesting that their expression is likely contributory to growth and cell immortalization in OPSCC [66]. In vitro, these mechanisms of E6 mediated transformation were dependent on the PSD-95/disc-large/Zo-1 (PDZ) binding motif, pointing towards potential targets for disease prevention and treatment [67].

Similarly, E7 promotes cell cycle progression in a multifactorial manner [68]. First, E7 blocks retinoblastoma protein (RB1), increasing production of transcription factors [69]. It also stimulates s-phase cyclins and acts on centrioles to rapidly progress cells through the DNA-amplification step of the cell cycle; oncoprotein (E6 and E7) expression has been associated with patterns of hypermethylation which promote E7 expressing genes, often via DNMT1s and maintained via miRNA dysregulation [70]. Inhibition of MAPK by E7 can also impede epithelial differentiation, contributing to mutagenesis. Together, E6 and E7 promote chromosome instability and inhibit apoptosis [71].

While these viral proteins are generally described as the drivers of mutagenesis, mouse studies of HPV have demonstrated that these oncoproteins alone are not sufficient for cellular transformation; rather, there are important synergistic steps, such as activation of h-RAS and expression of ErBb-2, required to achieve carcinogenesis [64,72]. This further reiterates that there are gaps in our knowledge as to how viral infection relates to other risk factors or predispositions for OPSCC.

### 3.3. Role of the Immune System in HPV-Mediated Oropharyngeal Carcinogenesis

Evasion of the immune system is equally important as cell cycle regulation in virally induced carcinogenesis [73]. Studies have demonstrated that tonsillar crypts express higher than normal cell surface ligand PD-L1, which negatively regulates programmed cell death [57]. Thus, when HPV infects this tissue, it adopts the tissue’s natural advantage (immune privilege) against the immune system. A similar phenomenon is observed in the cervical transformation zone, the primary site of HPV-mediated cervical cancer; the transformation zone houses fewer Langerhans’ cells, producing an immune privileged zone ideal for viral persistence [74]. Early in the viral lifecycle, the E5 protein modulates expression of the E6 and E7 genes via EGFR, which mitigates risk of identification by the host immune system [43,75,76]. Additionally, studies of benign HPV diseases, specifically recurrent respiratory papillomatosis (RRP), have demonstrated E5-mediated downregulation of human leukocyte antigens [24]. Similarly, there is evidence of muted T-helper cell responses in RRP and condyloma acuminate [62]. In high-risk HPV, E5 downregulates other cell surface markers that would activate otherwise an immune response. These examples could be indicative of how HPV subtypes persist in chronic infections [77].

Multiple additional mechanisms help HPV avoid the immune system [78]. Infection of the basal layer establishes a reservoir of infected cells. The viral lifecycle has adapted to produce the majority viral proteins once in the apical layers of epithelium, which do not have as many active immune cells [79]. In the intermediate layer, the non-lytic replicative cycle produces fewer free viral markers that would otherwise alert the innate immune system [80]. By avoiding activation of dendritic cells or other antigen presenting cells, an antigen-specific lymphocytic response is avoided, and the virus may persist. Toll-like receptors (TLRs) also play a role in recognizing common molecular patterns associated with pathogens. HPV proteins have been demonstrated to down regulate specific TLR, further muting the immune response to infection [81]. HPV-positive OPSCC have lower expression of TLR 9 and TLR 5 than HPV-negative OPSCC [82], which is important as impairment of the innate immune system negatively impacts a functional adaptive immune response [83]. Further, multiple studies investigating immune response in women with active HPV16 infection and/or cervical intraepithelial neoplasia (CIN) have demonstrated associations between HPV infection and weak adaptive immune response suggesting either a predisposition to infection or indicating there are potentially more mechanisms of immune evasion that are not well understood at this time. Though these data are specific to cervical HPV infection, the basic mechanisms of immune response to HPV may provide insight into the natural history of disease in other microenvironments, such as the oral cavity and pharynx [74].

Despite these immune-evading mechanisms, patients with HPV-positive OPSCC have a similar or better prognosis than those with HPV-negative cancer [84,85,86,87]. Phenotypically, HPV-positive tumors are distinct from HPV-negative tumors by their location, histology, and cell markers [88]. HPV-positive tumors are more likely to arise in the tonsils and typically exhibit nonkeratinizing patterns with abundant p16 antibodies [89]. Conversely, HPV-negative tumors are more likely keratinizing and commonly lose the p16 checkpoint. These variations may be inherent to the tissue that HPV readily infects; however, it is significant to detection and management of the disease. For example, p16 staining is commonly utilized as a surrogate for HPV testing [90]. HPV-positive tumors have also been demonstrated to have higher infiltration of lymphocytes than HPV-negative tumors, some of which (CD3+ and CD8+) were linked to increased survival [87,91]. There are also therapeutic distinctions between the two, with HPV-positive tumors typically responding better to treatment than HPV-negative tumors [54]. Patient age, stage at diagnosis, and other factors may confound this theory as the younger demographic afflicted with HPV-related OPSCC may tolerate treatment better than older patients—thus producing better outcomes [92,93]. There are some exceptions to this generalization, as well. Immunodeficiency, which may predispose patients to persistent HPV infection and carcinogenesis, may indicate worse prognosis in the setting of OPSCC, as it does in other HPV related cancers [16,94,95,96]. Additionally, exposure to other risk factors in HPV-positive patients, such as chronic tobacco use, nullifies this benefit [97]. This is thought to be due to baseline chromosomal instability in patients exposed to tobacco products than those who are not [98]. Though the field is beginning to consider the role of the oral microbiome in the pathogenesis of oral cavity cancer, to our knowledge the microbiome has not yet been examined as a factor related to HPV positive OPSCC [99].

### 3.4. The Microbiome as a Primer for Inflammation and HPV Mediated Tumor Progression

An important contributor of chronic mucosal inflammation is the microbiome [100,101,102,103,104,105,106,107,108,109,110]. Recent data show that oral microbial dysbiosis, with enrichment of *Fusobacterium* species, is associated with advanced oral cavity cancer stage; however, how this dysbiosis contributes to tumor progression remains an open question [111]. Most studies in this realm have focused on the oral microbiome and oral cavity cancer [112,113,114,115,116]. Our systematic-style search revealed a dearth of studies on the relationship between the microbiome and the two different types of OPSCC, namely, HPV positive and negative tumors. In cervicovaginal studies, increased species richness of the vaginal flora was linked to higher prevalence of HPV infection [117,118]. Attempts to replicate those studies in the oropharynx identified greater abundances of *Actinomycetaceae, Prevotellaceae, Veillonellaceae, Campylobacteraceae,* and *Bacteroidetes* in patients with HPV compared to those without HPV [118]. In these studies, *Fusobacteria* were lower in the setting of HPV infection, suggesting that microbes may be involved differently at various stages of disease progression. *Actinomyces* is commonly associated with periodontal disease, which is a risk factor for HNSCC [119]. The association between *Actinomyces*, periodontal disease, and HPV suggests that the inflammation and microabrasions associated with periodontal disease creates an environment primed for successful HPV infection [120]. One study identified microbial shifts correlated with shifts in gene expression that may contribute to carcinogenesis either directly (through virulence factors) or indirectly (through oxidative stress) [121]. Another study identified *Helicobacter pylori* co-infection is common in OPSCC, perhaps more so than HPV [122]. We suspect that tobacco use may facilitate a similar environment, explaining why cigarette smoking is a negative prognostic factor, even in HPV positive cancers [123,124]. In order to understand how microbiome compositional shifts can affect the mucosal homeostasis, a few groups have reported in silico gene expression analysis of the oral microbiome and how it relates to HPV positive OPSCC [125,126,127]. However, these in silico results have not been validated. Hence, understanding how the microbiome compositional shift of mucosa directly affects either the promotion or suppression of HPV cancer progression is very much still an open question.

### 3.5. Epstein–Barr Virus and Nasopharyngeal Carcinogenesis

Epstein–Barr virus (EBV) is a double stranded DNA virus in the herpesvirus family [128]. It was discovered by Dr. Epstein and his assistant, Ms. Barr, during a global investigation into the cause of Burkitt’s lymphoma in the 1960s [129]. Since then, in addition to Burkitt’s lymphoma, EBV has been definitively linked to several other cancers, including gastric carcinoma, t-cell lymphoma, and nasopharyngeal carcinoma (NPC) [130]. NPC is a subtype of squamous cell carcinoma born from the epithelium of the nasopharynx (the tubular passage posterior to the nasal cavity that connects inferiorly to the oropharynx) [131]. The most common site for NPC is the fossa of Rosenmüller, the pharyngeal recess located behind the torus tubarius (the medial cartilaginous end of the Eustachian tube). Given this location, symptoms of early-stage NPC seem non-specific and include congestion, headache, aural fullness, and/or mild epistaxis [132]. Due to the vague qualities of these symptoms, NPC is often diagnosed at later stages. Hence, the most common presentation of NPC is a painless neck mass—the result of regional cervical lymphatic spread [133]. Following careful workup and staging, treatment typically consists of radiotherapy with or without chemotherapy [134]. Although patients with early-stage disease have good outcomes with radiotherapy alone (87–93% 5-year survival), more intensive treatment strategies combining radiotherapy with chemotherapy are required to manage later-stage disease (63–81% 5-year survival) [54].

While EBV typically infects B- cells (hence its discovery in Burkitt’s lymphoma), evidence has demonstrated susceptibility for infection in multiple cell types in the human body [135]. It was previously believed that epithelial EBV infection, as opposed to B-cell infection, was lytic and time limited. However, viral markers of chronic EBV infection are seemingly ubiquitous across NPC subtypes, suggesting latent infection plays a critical role in carcinogenesis of the nasopharyngeal epithelium [136]. Likewise, recent research has identified chronic markers of EBV infection in some demyelinated lesions from patients with multiple sclerosis [137]. Though not all mechanisms for various disease states associated with EBV have been elucidated, connections between EBV and NPC have been closely investigated since the virus’ discovery. NPC was originally sought as a control cancer type to compare against Burkitt’s lymphoma. However, sera from NPC patients had large amounts of EBV antibodies. A 1995 study demonstrated clonal populations of EBV infected cells in NPC samples, further suggesting a role in immortalization of cells [138]. We now know that rich lymphoid tissue in the nasopharynx makes it an ideal reservoir for EBV. Though there remains some debate as to whether NPC predisposes to EBV infection or the opposite, a developing understanding of EBV’s lifecycle seems to support that viral proteins play a role in cellular transformation and accelerate tumorigenesis (Figure 3).

Additional risk factors for NPC include regular consumption of salted fish and Southeast Asian heritage. NPC develops significantly more frequently in patients living in southern China than in other regions of the world—perhaps due to certain human leukocyte antigen (HLA) profiles [136]. Genetic deletions on chromosome three, which impairs cells innate tumor suppression, may also predispose to NPC [143]. Taken together, this supports a genetic predisposition to NPC. However, historical migration patterns and modern epigenetic studies present a compelling argument that environmental exposures play an equally important role in tumorigenesis [144]. Other molecular events suspected to facilitate latent infection, and thus contribute to carcinogenesis, include hypermethylation of specific genes (e.g., RASSF1A and BLU), deactivation of p16, deactivation of lactoferrin, and upregulation of cyclin D1 production [13,145,146].

### 3.6. Cellular Mechanisms of EBV-Mediated Nasopharyngeal Carcinogenesis

During EBV’s dormant phase, it integrates into host cell DNA to increase expression of latent viral proteins and modifies host methylation patterns [128]. EBV-derived latent membrane proteins (LMPs) are a key driver of clonal proliferation and transformation of EBV-infected epithelium (Figure 3). More specifically, LMP1 is thought to be crucial to evasion of programed cell-death and, thus, necessary for malignant transformation of EBV infected cells [147,148,149]. Among its mechanisms for prolonging cell survival, LMP1 increases telomerase activity via human telomerase reverse transcriptase (hTERT) [150,151,152]. It also activates c-myc and the ATR/Chk1 pathway, progressing the cell cycle and aiding in the hyperproliferative phase of infection [153]. LMP1-mediated inactivation of AMPK has been demonstrated to promote continued growth of infected cells [154]. Special AT-rich binding protein 1 (SATB1), a poorly understood transcription factor implicated in cancers such as breast and gastric, is directly regulated by LMP1 in EBV infected epithelial cells. SATB1 has proliferative and anti-apoptotic properties through upregulation of more downstream factors, such as Survivin [155]. This protein is also directly targeted by LMP1, further demonstrating the multifactorial mechanisms in which EBV promotes immortality [156]. While the majority of protein targets regulate cell cycle regulation, other hypothesized targets facilitate angiogenesis, transformation, and tumor spread [149,157].

Once EBV is integrated into cells, LMP2 facilitates cell migration, contributing to cancer spread and metastasis [158]. Specifically, LMP2A has been directly correlated with integrin ITGα6, which has been correlated with cellular motility and invasiveness of cancer. Furthermore, LMP2A initiates an epithelial to mesenchymal transformation that increases motility of cells. LMP2A has also been correlated with presence and increasing size of side populations of cells that possess characteristics of stem cells (e.g., self-propagation). The cancer stem cell theory, stating that tumorigenesis and recurrence are the result of cancerous stem-cell-like cells (CSCs), is reinforced by evidence that LMP1 and LMP2A can activate the hedgehog and NOTCH pathways [159].

These data are valuable because identification of early cellular changes in NPC pathogenesis may serve clinicians in early identification for disease and/or in guiding treatment. One study found that higher levels of LMP1 were associated with larger tumor size and poorer 5-year survival [160]. Others have identified EBV DNA, and downstream effects of viral proteins (such as markers of oxidative stress, RIP3 hypermethylation) to be informative prognostic indicators [161]. Understanding the relationship between baseline risk, EBV infection, and inflammation may help us to explore disease targeted therapies for NPC.

### 3.7. Role of the Immune System in EBV-Mediated Nasopharyngeal Carcinogenesis

Because EBV is typically acquired early in life, the adaptive immune system is primed to recognize viral proteins upon reactivation of disease. Simultaneously, pathogen-associated molecular patterns are recognized by pattern recognition receptors at the cell’s surface and trigger an inflammatory response. Downstream impacts of activating inflammatory NF*k*B and STAT3 pathways increase cytokines, chemokines, and adhesion molecules [162,163]. While this inflammation is aimed at generating a lethal environment for pathogens, it paradoxically promotes immortalization of the host cell, as described above [164]. Some infected cells with LMP-mediated NF*k*B activation have expressed Id1, which negatively regulates p16 further amplifying replication and immortalization [162]. Other studies demonstrate activation of NADPH oxidases [165]. This is significant because the resulting reactive oxygen species can develop radioresistance in cancer cells, thus hindering efficacy of treatment.

LMP1 is the key immunomodulatory protein produced by EBV. LMP1 evades cell death by suppressing the natural tumor induced necroptosis signaling via a variety of mechanisms [141]. First, LMP1 hypermethylates the gene regions for RIP3, the kinase responsible to programmed cell death, as noted above [163]. Another major tumor suppressor gene, TGFBR3, resides on chromosome 3, which is often deleted in EBV infected cells. EBV proteins generate somatic changes to maintain constitutive activation of NF*k*B without alarming the natural immune response [140]. These changes include modification of NF*k*B regulators and amplification of cell surface proteins such as LTBR, which further mutes the immune response [166]. EBV-encoded small RNA (EBER) are known to trigger an immune response via type I interferons [167]. One study also demonstrated associations between EBV infection and CD1d expression, which is necessary for a natural killer T cell response [168]. Taken together, these mechanisms help promote cell division while conferring a pro-inflammatory niche.

### 3.8. Does the Microbiome Play a Role in Nasopharyngeal Carcinogenesis?

There are no studies examining the interplay of the pharyngeal microbiome, EBV, and carcinogenesis, despite obvious pattens of NPC worldwide. This knowledge gap may be attributable to the general rarity of nasopharyngeal carcinoma as a whole and its concentration in less affluent countries. Some research suggests endemic regions may be at risk for a more virulent strain of EBV than other regions [144]. However, even so, the incidence of NPC is approximately 0.5–25 per 100,000 males in southern China (the most severely impacted subpopulation) whereas EBV infection is virtually ubiquitous [144]. This leads one to ponder what factors could contribute to the discrepancy in EBV infection and nasopharyngeal carcinogenesis. Occupational exposures, exhausts, smoke, and caustic agents are also associated with increased risk of NPC, both in endemic regions and otherwise [169]. Diet and air quality have also been linked to disease incidence [144]. As we know from the literature on HPV and OPSCC, all of these factors can impact the microbiome. Thus, it is logical to propose that the pharyngeal microbiome could contribute to chronic cellular stress and mutagenesis in the setting of EBV-mediated carcinogenesis. Clearly, more research is warranted in this domain.

## 4. Discussion

The purpose of this review was to critically assess whether the pharyngeal microbiome and viral infection synergistically induce carcinogenesis by inducing a chronic inflammatory state. The literature clearly demonstrates that viruses (EBV and HPV) mediate cell cycle dysregulation and host inflammation. However, there is scant literature on the interplay between these oncogenic viruses and the local microbiome. It is well established that, in both HPV and EBV oncogenic pathways, inflammation from prior environmental exposures, including smoke and/or chemicals, increases patients’ risk for carcinogenesis. This idea supports the theory that long-term, low-grade inflammation secondary to chronic infection may also promote mutagenesis.

Perturbations in the microbiome have been linked to high morbidity diseases including diabetes, poor maternal-fetal outcomes, and cancer [2]. It is vital that we better characterize the mechanisms by which microbial dysbiosis promotes disease and how disease states impact microbial homeostasis. Evidence suggests that periodontal disease, shifts in commensal bacteria populations, and associated inflammation may lower the threshold for HPV infection and prime cells for pathogen-mediated mutagenesis [120]. Outcomes in EBV-mediated NPC suggest that inflammation may play a similar role in this setting, though consideration of the microbiome’s role has been limited.

In both cases, the pharynx provides a unique and complex microenvironment for carcinogenesis. Tonsillar tissue, with its close approximation of epithelium and lymphoid tissue, is an ideal reservoir for latent viral infections [170]. Adenoid-derived B lymphocytes more rapidly and efficiently uptake EBV infection than B cells from anywhere else in the body [171]. Moreover, the viruses passively utilize natural advantages of the pharyngeal tissue, such as moderate immune privilege, to propagate undetected, while also actively modifying important cell cycle regulators. One study demonstrated that coinfection of these tissues with HPV and EBV significantly increased invasiveness of tonsillar and base of tongue cancer [172].

With the expanding fund of knowledge related to virally induced carcinogenesis in the pharynx, we must consider how can these lessons can be applied in predicting or mitigating disease. Viral serology has been proposed as an adjunct for head and neck cancer screening [173,174]. However, given that the vast majority of individuals with viral exposure do not develop cancer, viral serology is more often used for prognostication and surveillance of disease [175]. From this perspective, it could be useful to study shifts in the microbiome to predict or detect disease. Similarly, though viral DNA is typically regarded as more sensitive and specific for detection, immune markers can be important for early detection and surveillance of infection. These may be useful for treatment planning, as well. For example, EBV associated NPC with high tumor infiltrating CD-8 cells have better prognosis and respond well to less invasive treatment [176].

Perhaps the most significant implication of tying viral infection to cancer development and the microbiome is the concept of preventing disease. Though there are some notable genetic predispositions, there are also confounding variables, such as environmental exposures, diet, and alcoholism, which may confound the impact of those data. All of these factors are known to alter the microbiome. As such, it could be possible to modulate the microbiome—either with probiotic augmentation, antibiotic modulation, or antiseptic elimination—to modulate the course of disease itself.

## 5. Conclusions

The incidence of head and neck cancer is growing globally, conferring high morbidity and mortality to affected populations. Chronic viral infection with HPV/EBV and the associated persistent low-grade inflammation may have additive mutagenic effects within infected tissue. The existing literature presents compelling evidence that HPV and EBV directly modulate cell cycle replication and tumor gene expression, contributing to inflammatory carcinogenesis in the pharynx. However, literature linking the pharyngeal microbiome to this process is lacking, which warrants further research. Knowledge of specific shifts in the microbiome (including viruses, bacteria, and fungi) could be applied to improve diagnostic workup, to stratify patient risk, and to drive the development of new treatment modalities.

## Figures and Tables

**Figure 1 cancers-14-03120-f001:**
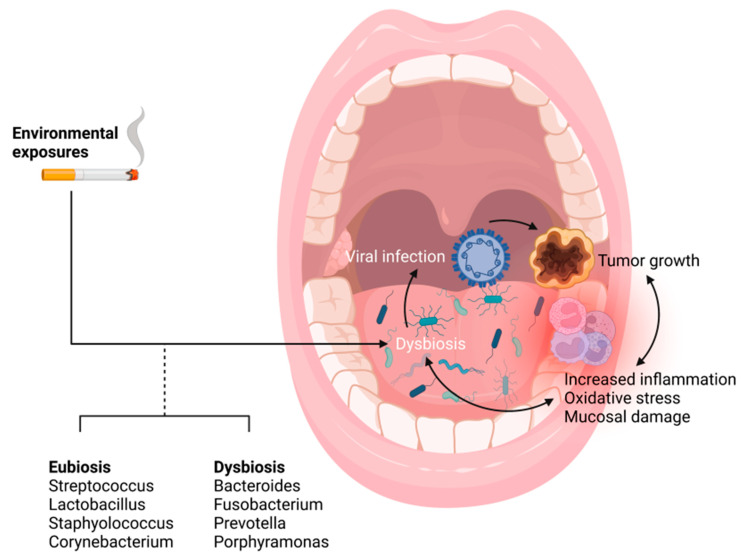
Overview of environment-host-microbiome-tumor interactions. Schematic overview of the interplay between the environment, the oral microbiome, human papillomavirus (HPV), a tonsillar tumor, and host inflammation. Environmental factors, such as tobacco smoke, alter the oral microbiome [7,25,26,27,28,29]. Meanwhile, bacterial shifts in the oral microbiome modulate viral proliferation and infection [30]. The interplay of these microbiological factors, a tumor, and the host immune system is complex [31,32,33]. Figure generated using BioRender.

**Figure 2 cancers-14-03120-f002:**
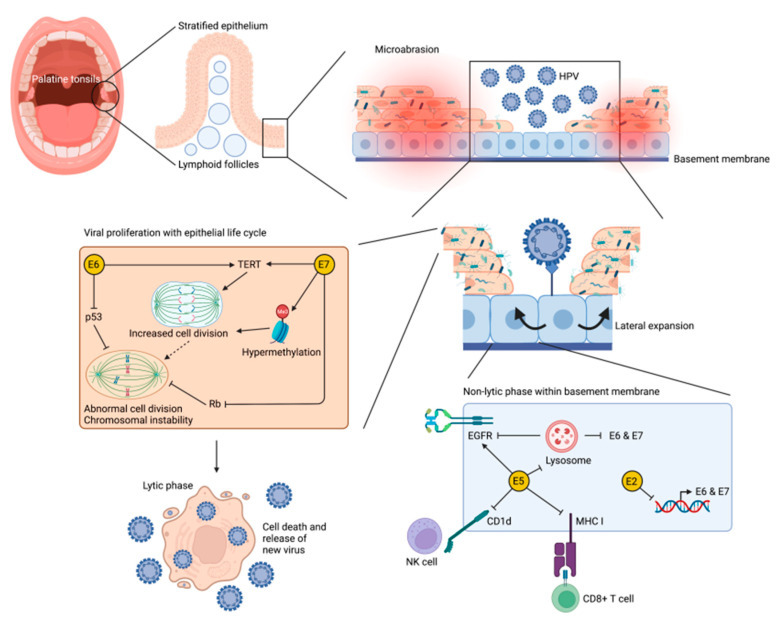
Interplay of HPV, the microbiome, and the oropharynx. Overview of the HPV lifecycle within a tonsillar tumor. Microabrasions in tonsillar epithelium lead to HPV infection of basilar and epithelial cells. Basilar cells undergo lateral expansion during the non-lytic phase of the viral cycle, primarily mediated by E5 [43]—which modulates EGFR signaling [44], downregulates CD1d [45], prevents MHCI trafficking [46], and regulates autophagy [47]. Epithelial cells undergo proliferation and transformation prior to the lytic phase of the virus’s life cycle, primarily mediated by E6 (which degrades p53) and E7 (which similarly degrades Rb) [48]. E6 and E7 also affect cell division via TERT and methylation [49,50]. Figure generated using BioRender.

**Figure 3 cancers-14-03120-f003:**
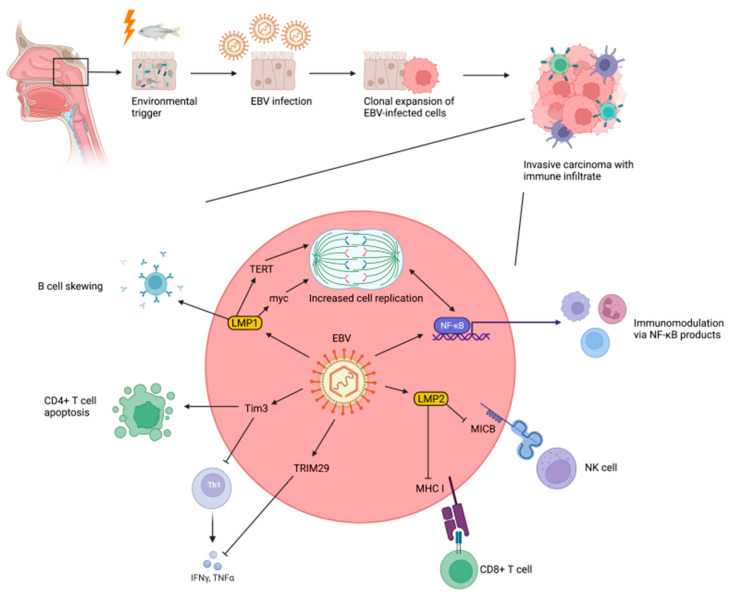
Growth and immune evasion of EBV-mediated NPC. Typical pathway for EBV infection and cellular transformation (**top**). Mechanisms of EBV-mediated cellular transformation and immune evasion (**bottom**). EBV alters immune checkpoints, including TIM-3 and TRIM29 [139]. The virus also alters the immune system via modulation of NFkB. [140]. EBV-specific products LMP1 [141] and LMP2 [142] also alter the cell cycle and aid in immune evasion [128]. Figure generated using BioRender.

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
