# Peer review of "It Takes Two to Tango: A Review of Oncogenic Virus and Host Microbiome Associated Inflammation in Head and Neck Cancer"

_cancers, 2022, doi:10.3390/cancers14133120_

Round 1

Reviewer 1 Report

This "review of oncogenic virus and host microbiome associated inflammation in head and neck cancer" is generally well written and interesting.  The manuscript provides a nice brief review of the oncogenic properties of HPV and EBV, and of their attendant manipulation of the immune system.  However, it has two significant weaknesses.  First, there is very little information presented on the microbiome or inflammation in the oropharynx and nasopharynx, and how they affect or are affected by HPV and EBV.  The authors acknowledge this, but the reader is left to wonder what the 120 papers found in the authors' systematic review of microbiome and inflammation-related HPV and EBV cancers reveal.  The authors refer to a much smaller subset:  the entire manuscript lists only 95 references that are clearly not all derived from the search.  A paper on the presence of H pylori and HPV in OPSCC, published over a month before the authors' final search, was not mentioned in this manuscript (Astl et al., Genome of Helicobacter pylori and Serotype of HPV Detected in Oropharyngeal and Laryngeal Cancer and Chronic Inflammation Patients. Int J Environ Res Public Health. 2021 Sep 10;18(18):9545).  Second, many statements of fact do not refer to sources for the information.  That is unacceptable.  A major revision of this manuscript, including many more references, a clear assessment of what the systematic review found, and a revision of the title to reflect the true content of the manuscript, could be a significant contribution to the field.

Specific comments:

Lines 104 and 109 suggest that at least 120 papers chosen to be reviewed and summarized; however, the number of references for this manuscript is only 95.  The remaining papers that were part of the systematic review should also be summarized in the text.  It would be helpful to provide a list of all the selected papers in a table (possibly as a supplement).

Line 107:  "complimentary databased":  both words need correction; should be "complementary databases"

Line 107:  PubMed should be listed here if it was also used for specific follow-up searches.  This paragraph leaves the impression that only the primary search was conducted in PubMed. 

Lines 109-110 should be added to the prior paragraph.

Lines 119-123:  This is a run-on sentence.  Also, reference 26, on the role of micro-RNAs in cervical cancer metastases, does not appear to be the best reference for this very general statement.  Many reviews have been written on this topic, in addition to the authors' (ref 27).

Line 129 (Figure legend):  Please list references for the information in the figure (references in the accompanying text are not sufficient).  References should cover all the information in the figure, including regulation of TERT, EGFR, MHC1, CD1d, chromatin hypermethylation etc.

Line 160:  Please provide a reference for the following statements:  "While dysplasia, as seen in premalignant lesions of the cervix, is not observed in the oropharynx" and "E5, E6, and E7, have been demonstrated to be key players in HPV positive OPSCC".  Ref 22 does not address oropharyngeal dysplasia or the importance of E6 and E7 in the oropharynx.

Line 166:  The word "by" seems to be missing from "perhaps manipulation"

Lines 203-204:  Please provide a reference for the following statement:  "Early in the viral lifecycle, the E5 protein acts to slow translation of the E6 and E7 genes"

Line 211:  "Multiple additional mechanisms help HPV avoid the immune system" (If you want to use "aid" you need to use the infinitive of avoid – "to avoid")

Lines 211-218:  The following two sentences virtually negate each other's importance:  

1)  Infection of the basal layer inherently ensures multiple layers of epithelial cells protecting virally infected cells from circulating immune cells

2)  The viral lifecycle has adapted to produce the majority viral proteins once in the apical layers of epithelium, which don’t have as many active immune cells.

The conclusion seems to be that no layers are completely safe from immune cells.  Please revise and strengthen your arguments, or remove.

Line 240:  "may confound this theory" – please clarify

Line 241:  "acceptions" – should be "exceptions"

Line 243: "as it does in other HPV related cancers" – please provide a reference

Lines 244-246:  "Additionally, exposure to other risk factors in HPV-positive patients, such as chronic tobacco use, nullifies this benefit. This is thought to be due to baseline chromosomal instability in patients exposed to tobacco products than those who are not." – please provide a reference.  (Reference 51 does not address tobacco and chromosomal instability.)

Lines 257-258:  "In cervicovaginal studies, increased species richness of the vaginal flora were linked to higher prevalence of HPV infection" – please provide a reference

Lines 269-270:  "Tobacco use may facilitate a similar environment, explaining why cigarette smoking is a negative prognostic factor, even in HPV positive cancers." – please provide a reference.  If there is no evidence that tobacco facilitates a similar environment, please remove.

Lines 278-289:  Please provide references for the following:  "Epstein-Barr virus (EBV) is a double stranded DNA virus in the herpesvirus family. It was discovered by Dr. Epstein and his assistant, Ms. Barr, during a global investigation into the cause of Burkitt’s lymphoma in the 1960s. Since then, in addition to Burkitt’s lymphoma, EBV has been definitively linked to several other cancers, including gastric carcinoma, t-cell lymphoma, and nasopharyngeal carcinoma (NPC). NPC is a subtype of squamous cell carcinoma born from the epithelium of the nasopharynx (the tubular passage posterior to the nasal cavity that connects inferiorly to the oropharynx). The most common site for NPC is the fossa of Rosenmüller, the pharyngeal recess located behind the torus tubarius (the medial cartilaginous end of the Eustachian tube). Given this location, symptoms of early-stage NPC seem non-specific and include congestion, headache, aural fullness, and/or mild epistaxis. Due to the vague qualities of these symptoms, NPC is often diagnosed at later stages. Hence, the most common presentation of NPC is a painless neck mass—the result of regional cervical lymphatic spread."

The above lack of references was egregious, and showed that unsupported statements were a major problem in this manuscript.  I did not continue to list individual instances of this problem. 

Author Response

Reviewer 1

Comments and Suggestions for Authors

This "review of oncogenic virus and host microbiome associated inflammation in head and neck cancer" is generally well written and interesting.  The manuscript provides a nice brief review of the oncogenic properties of HPV and EBV, and of their attendant manipulation of the immune system.  However, it has two significant weaknesses.  First, there is very little information presented on the microbiome or inflammation in the oropharynx and nasopharynx, and how they affect or are affected by HPV and EBV.  The authors acknowledge this, but the reader is left to wonder what the 120 papers found in the authors' systematic review of microbiome and inflammation-related HPV and EBV cancers reveal.  The authors refer to a much smaller subset:  the entire manuscript lists only 95 references that are clearly not all derived from the search.  A paper on the presence of H pylori and HPV in OPSCC, published over a month before the authors' final search, was not mentioned in this manuscript (Astl et al., Genome of Helicobacter pylori and Serotype of HPV Detected in Oropharyngeal and Laryngeal Cancer and Chronic Inflammation Patients. Int J Environ Res Public Health. 2021 Sep 10;18(18):9545).  Second, many statements of fact do not refer to sources for the information.  That is unacceptable.  A major revision of this manuscript, including many more references, a clear assessment of what the systematic review found, and a revision of the title to reflect the true content of the manuscript, could be a significant contribution to the field.

We thank the reviewer for their detailed review of this manuscript, which has undoubtedly made this work stronger. One of the reviewer’s main criticisms had to do with the scarce information linking the microbiome, inflammation, and head and neck cancer. This is indeed true; there isn’t much literature on the subject. Part of the goal of this review was to attempt to uncover articles linking these fields and ideas. We approached this with a systematic-style search, as detailed in the methods. The goal was not to perform a systematic review, per se; but, rather, to cast a wide net in order to find articles linking all of these subjects. No search method is perfect; and, as highlighted by the reviewer’s reference above, some articles were not found by the search. In the revised manuscript, we have included the specific reference mentioned by the reviewer, and have done our best to incorporate many others, throughout the manuscript.

Along those lines, the reviewer’s other major criticisms had to do with a lack of citations. This indeed can be problematic. In reviewing our work, we noticed that many of these missing citations followed our clinical expertise and opinion. In fact, the paragraph noted to be most “egregious,” was nearly entirely clinical in nature. While clinical opinion can often go uncited, we have added citations throughout to remove any and all doubts. We have gone beyond and poured through the whole manuscript while adding citations most everywhere to address the reviewer’s concerns.

Specific comments:

Lines 104 and 109 suggest that at least 120 papers chosen to be reviewed and summarized; however, the number of references for this manuscript is only 95.  The remaining papers that were part of the systematic review should also be summarized in the text.  It would be helpful to provide a list of all the selected papers in a table (possibly as a supplement).

As noted above, the goal of the systematic-style search was to identify studies (linking the inflammation, the microbiome, EBV/HPV, and head and neck cancer) that we may not have otherwise found via a more traditional (narrative) style review. We were not aiming to perform a true systematic review; and, had we done so, we indeed would have included 1) more detailed methods, 2) a list of all identified papers, and 3) a PRISMA diagram (as per Reviewer 3). However, we agree with the merit of your point, and in the revised manuscript, the text in the methods section has now been updated to clarify these differences and to add more detail:

In order to identify studies linking the inflammation, the microbiome, EBV/HPV, and head and neck cancer, we conducted a systematic-style search in PubMed. The terms pharynx, oropharynx, nasopharynx (and related anatomy), viral infections, inflammation, EBV, HPV, and carcinogenesis were queried. The specific search query was: ("Pharynx"[Mesh:NoExp] OR "Oropharynx"[Mesh] OR "Nasopharynx"[Mesh] OR "Palate, Soft"[Mesh] OR "Eustachian Tube"[Mesh]) AND("Viruses"[Mesh] OR "Virology"[Mesh] OR "Virome"[Mesh] OR "Infections"[Mesh] OR "Inflammation"[Mesh] OR "Epstein-Barr Virus Infections"[Mesh] OR "Papillomaviridae"[Mesh] "Epstein Barr"[tiab] OR "EBV infection"[tiab] OR "EBV infections"[tiab] OR papillomaviridae[tiab] OR "human papilloma virus"[tiab] OR "human papilloma viruses"[tiab] OR "human papillomavirus"[tiab] OR HPV[tiab]) AND ("Carcinogenesis"[Mesh] OR "Oropharyngeal Neoplasms"[Mesh] OR "Nasopharyngeal Neoplasms"[Mesh]). No date limits were imposed, and the last search was performed on October 21st, 2021.

Our search initially resulted in 248 records. Titles were then independently evaluated for relevance by authors JG and MGM (using a citation manager), resulting in 114 relevant abstracts. Whole manuscripts were further screened for relevance. Non-original articles, review articles, non-human studies, and abstract-only publications were then excluded. Additionally, studies published in languages other than English were excluded. Interrater conflicts were assessed and re-designated relevant or irrelevant by MGM, leaving 45 articles for review. Any pertinent topic areas that were uncovered following this thorough search were supplemented with specific searches in complementary databases, such as Google Scholar, PubMed, and Web of Science. Additional searches to specifically link the microbiome, HPV, and OPSCC yielded 6 pertinent articles; no articles were found linking the microbiome, EBV, and NPC

Line 107:  "complimentary databased":  both words need correction; should be "complementary databases"

Thank you for identifying these typos. The words have been corrected.

Line 107:  PubMed should be listed here if it was also used for specific follow-up searches.  This paragraph leaves the impression that only the primary search was conducted in PubMed.

Thank you for noting the lack of clarity on our part. In the revised manuscript, we have modified the text to include PubMed.

Lines 109-110 should be added to the prior paragraph.

Thank you! We have corrected this suggested formatting.

Lines 119-123:  This is a run-on sentence.  Also, reference 26, on the role of micro-RNAs in cervical cancer metastases, does not appear to be the best reference for this very general statement.  Many reviews have been written on this topic, in addition to the authors' (ref 27).

We appreciate your keen eye and have broken the run-on into two sentences. Though the reference is specific to microRNAs, the authors of the referenced article provide an excellent summation of the viral life cycle of HPV and the epidemiology of its associated disease states; thus, we have kept it as a citation here. To further emphasize the topic, we have added additional citations by other groups. The revised manuscript now reads:

Most infections are cleared within six months to a year; however, persistent infection with high-risk HPV is the one of the primary etiologic factors for cervical cancer, as well as other anogenital and head and neck cancers[10,16,28,29]. Specifically, about 70% of these cancers are caused by HPV16 and HPV18 genotypes[10,11,30,31]; these strains are detected significantly more often in biopsies of malignant tonsillar disease than in benign tonsillar pathology[32,33].

  1. Marur, S.; D’Souza, G.; Westra, W.H.; Forastiere, A.A. HPV-Associated Head and Neck Cancer: A Virus-Related Cancer Epidemic. Lancet Oncol 2010, 11, 781–789, doi:10.1016/s1470-2045(10)70017-6.
  2. Gillison, M.L. Human Papillomavirus-Related Diseases: Oropharynx Cancers and Potential Implications for Adolescent HPV Vaccination. J Adolescent Health 2008, 43, S52–S60, doi:10.1016/j.jadohealth.2008.07.002.
  3. Hausen, H. zur Papillomaviruses and Cancer: From Basic Studies to Clinical Application. Nat Rev Cancer 2002, 2, 342–350, doi:10.1038/nrc798.
  4. Szymonowicz, K.A.; Chen, J. Biological and Clinical Aspects of HPV-Related Cancers. Cancer Biology Medicine 2020, 17, 864–878, doi:10.20892/j.issn.2095-3941.2020.0370.
  5. El-Mofty, S.K. Human Papillomavirus (HPV) Related Carcinomas of the Upper Aerodigestive Tract. Head Neck Pathology 2007, 1, 181–185, doi:10.1007/s12105-007-0021-6.
  6. Santos, J.M.O.; Silva, S.P. da; Costa, N.R.; Costa, R.M.G. da; Medeiros, R. The Role of MicroRNAs in the Metastatic Process of High-Risk HPV-Induced Cancers. Cancers 2018, 10, 493, doi:10.3390/cancers10120493.
  7. Gupta, S.; Kumar, P.; Das, B.C. HPV: Molecular Pathways and Targets. Curr Prob Cancer 2018, 42, 161–174, doi:10.1016/j.currproblcancer.2018.03.003.
  8. Ilmarinen, T.; Munne, P.; Hagström, J.; Haglund, C.; Auvinen, E.; Virtanen, E.I.; Haesevoets, A.; Speel, E.J.M.; Aaltonen, L.-M. Prevalence of High-Risk Human Papillomavirus Infection and Cancer Gene Mutations in Nonmalignant Tonsils. Oral Oncol 2017, 73, 77–82, doi:10.1016/j.oraloncology.2017.08.010.
  9. Herberhold, S.; Hellmich, M.; Panning, M.; Bartok, E.; Silling, S.; Akgül, B.; Wieland, U. Human Polyomavirus and Human Papillomavirus Prevalence and Viral Load in Non-Malignant Tonsillar Tissue and Tonsillar Carcinoma. Med Microbiol Immun 2016, 206, 93–103, doi:10.1007/s00430-016-0486-6.

Line 129 (Figure legend):  Please list references for the information in the figure (references in the accompanying text are not sufficient).  References should cover all the information in the figure, including regulation of TERT, EGFR, MHC1, CD1d, chromatin hypermethylation etc.

Thank you for your suggestion. The figure legends all have been adjusted to include all pertinent citations in the revised manuscript:

Figure 1: overview of environment-host-microbiome-tumor interactions

Schematic overview of the interplay between the environment, the oral microbiome, human papillomavirus (HPV), a tonsillar tumor, and host inflammation. Environmental factors, such as tobacco smoke, alter the oral microbiome[156–158]. Meanwhile, bacterial shifts in the oral microbiome modulate viral proliferation and infection[159]. The interplay of these microbiological factors, a tumor, and the host immune system is complex[160–162]. Figure generated using BioRender.

  1. Colman, G.; Beighton, D.; Chalk, A.J.; Wake, S. Cigarette Smoking and the Microbial Flora of the Mouth*. Aust Dent J 1976, 21, 111–118, doi:10.1111/j.1834-7819.1976.tb02833.x.
  2. Morris, A.; Beck, J.M.; Schloss, P.D.; Campbell, T.B.; Crothers, K.; Curtis, J.L.; Flores, S.C.; Fontenot, A.P.; Ghedin, E.; Huang, L.; et al. Comparison of the Respiratory Microbiome in Healthy Nonsmokers and Smokers. Am J Resp Crit Care 2013, 187, 1067–1075, doi:10.1164/rccm.201210-1913oc.
  3. Mason, M.R.; Preshaw, P.M.; Nagaraja, H.N.; Dabdoub, S.M.; Rahman, A.; Kumar, P.S. The Subgingival Microbiome of Clinically Healthy Current and Never Smokers. Isme J 2015, 9, 268–272, doi:10.1038/ismej.2014.114.
  4. Tada, A.; Senpuku, H. The Impact of Oral Health on Respiratory Viral Infection. Dent J 2021, 9, 43, doi:10.3390/dj9040043.
  5. Shetty, S.S.; Padam, K.S.R.; Hunter, K.D.; Kudva, A.; Radhakrishnan, R. Biological Implications of the Immune Factors in the Tumour Microenvironment of Oral Cancer. Arch Oral Biol 2021, 133, 105294, doi:10.1016/j.archoralbio.2021.105294.
  6. Zong, Y.; Zhou, Y.; Liao, B.; Liao, M.; Shi, Y.; Wei, Y.; Huang, Y.; Zhou, X.; Cheng, L.; Ren, B. The Interaction Between the Microbiome and Tumors. Front Cell Infect Mi 2021, 11, 673724, doi:10.3389/fcimb.2021.673724.
  7. Kurago, Z.; Loveless, J. Microbial Colonization and Inflammation as Potential Contributors to the Lack of Therapeutic Success in Oral Squamous Cell Carcinoma. Frontiers Oral Heal 2021, 2, 739499, doi:10.3389/froh.2021.739499.

Figure 2: interplay of HPV, the microbiome, and the oropharynx

Overview of the HPV lifecycle within a tonsillar tumor. Microabrasions in tonsillar epithelium lead to HPV infection of basilar and epithelial cells. Basilar cells undergo lateral expansion during the non-lytic phase of the viral cycle, primarily mediated by E5[43]—which modulates EGFR signaling[163], downregulates CD1d[45], prevents MHCI trafficking[164], and regulates autophagy[165]. Epithelial cells undergo proliferation and transformation prior to the lytic phase of the virus’s life cycle, primarily mediated by E6 (which degrades p53) and E7 (which similarly degrades Rb)[166]. E6 and E7 also affect cell division via TERT and methylation[167,168]. Figure generated using BioRender.

  1. Venuti, A.; Paolini, F.; Nasir, L.; Corteggio, A.; Roperto, S.; Campo, M.S.; Borzacchiello, G. Papillomavirus E5: The Smallest Oncoprotein with Many Functions. Mol Cancer 2011, 10, 140–140, doi:10.1186/1476-4598-10-140.
  2. Crusius, K.; Auvinen, E.; Steuer, B.; Gaissert, H.; Alonso, A. The Human Papillomavirus Type 16 E5-Protein Modulates Ligand-Dependent Activation of the EGF Receptor Family in the Human Epithelial Cell Line HaCaT. Exp Cell Res 1998, 241, 76–83, doi:10.1006/excr.1998.4024.
  3. Miura, S.; Kawana, K.; Schust, D.J.; Fujii, T.; Yokoyama, T.; Iwasawa, Y.; Nagamatsu, T.; Adachi, K.; Tomio, A.; Tomio, K.; et al. CD1d, a Sentinel Molecule Bridging Innate and Adaptive Immunity, Is Downregulated by the Human Papillomavirus (HPV) E5 Protein: A Possible Mechanism for Immune Evasion by HPV. J Virol 2010, 84, 11614–11623, doi:10.1128/jvi.01053-10.
  4. Marchetti, B.; Ashrafi, G.H.; Tsirimonaki, E.; O’Brien, P.M.; Campo, M.S. The Bovine Papillomavirus Oncoprotein E5 Retains MHC Class I Molecules in the Golgi Apparatus and Prevents Their Transport to the Cell Surface. Oncogene 2002, 21, 7808–7816, doi:10.1038/sj.onc.1205885.
  5. Aranda‐Rivera, A.K.; Cruz‐Gregorio, A.; Briones‐Herrera, A.; Pedraza‐Chaverri, J. Regulation of Autophagy by High‐ and Low‐risk Human Papillomaviruses. Rev Med Virol 2021, 31, e2169, doi:10.1002/rmv.2169.
  6. Hoppe-Seyler, K.; Bossler, F.; Braun, J.A.; Herrmann, A.L.; Hoppe-Seyler, F. The HPV E6/E7 Oncogenes: Key Factors for Viral Carcinogenesis and Therapeutic Targets. Trends Microbiol 2018, 26, 158–168, doi:10.1016/j.tim.2017.07.007.
  7. Doorslaer, K.V.; Burk, R.D. Association between HTERT Activation by HPV E6 Proteins and Oncogenic Risk. Virology 2012, 433, 216–219, doi:10.1016/j.virol.2012.08.006.
  8. Burgers, W.A.; Blanchon, L.; Pradhan, S.; Launoit, Y. de; Kouzarides, T.; Fuks, F. Viral Oncoproteins Target the DNA Methyltransferases. Oncogene 2007, 26, 1650–1655, doi:10.1038/sj.onc.1209950.

Figure 3: growth and immune evasion of EBV-mediated NPC

Typical pathway for EBV infection and cellular transformation (top). Mechanisms of EBV-mediated cellular transformation and immune evasion (bottom). EBV alters immune checkpoints, including TIM-3 and TRIM29[169]. The virus also alters the immune system via modulation of NFkB.[144]. EBV-specific products LMP1[143] and LMP2[170] also alter the cell cycle and aid in immune evasion[109]. Figure generated using BioRender.

  1. Biggi, A.F.B.; Oliveira, D.E. de The Epstein-Barr Virus Hacks Immune Checkpoints: Evidence and Consequences for Lymphoproliferative Disorders and Cancers. Biomol 2022, 12, 397, doi:10.3390/biom12030397.
  2. Senten, J.R. van; Fan, T.S.; Siderius, M.; Smit, M.J. Viral G Protein-Coupled Receptors as Modulators of Cancer Hallmarks. Pharmacol Res 2020, 156, 104804, doi:10.1016/j.phrs.2020.104804.
  3. Wang, L.; Ning, S. New Look of EBV LMP1 Signaling Landscape. Cancers 2021, 13, 5451, doi:10.3390/cancers13215451.
  4. Cen, O.; Longnecker, R. Epstein Barr Virus Volume 2, One Herpes Virus: Many Diseases. Curr Top Microbiol 2015, 391, 151–180, doi:10.1007/978-3-319-22834-1_5.
  5. Bauer, M.; Jasinski-Bergner, S.; Mandelboim, O.; Wickenhauser, C.; Seliger, B. Epstein–Barr Virus—Associated Malignancies and Immune Escape: The Role of the Tumor Microenvironment and Tumor Cell Evasion Strategies. Cancers 2021, 13, 5189, doi:10.3390/cancers13205189.

Line 160:  Please provide a reference for the following statements:  "While dysplasia, as seen in premalignant lesions of the cervix, is not observed in the oropharynx" and "E5, E6, and E7, have been demonstrated to be key players in HPV positive OPSCC".  Ref 22 does not address oropharyngeal dysplasia or the importance of E6 and E7 in the oropharynx.

Thank you! In the cited article, it is stated that “...dysplasia prior to malignant transformation has not been detected, making a screening program difficult.” To bolster the fact that these viral proteins are key in the pathophysiology for both cervical and oral cavity infections, we have added another reference (Andersen et al.), which discusses the roles of viral proteins.

Line 166:  The word "by" seems to be missing from "perhaps manipulation"

Thank you for the keen eye. We have corrected this in the revised manuscript as suggested

Lines 203-204:  Please provide a reference for the following statement:  "Early in the viral lifecycle, the E5 protein acts to slow translation of the E6 and E7 genes"

Thank you! We have now added several citations supporting the fact that E5 modulates protein product of E6 and E7:

  1. Venuti, A.; Paolini, F.; Nasir, L.; Corteggio, A.; Roperto, S.; Campo, M.S.; Borzacchiello, G. Papillomavirus E5: The Smallest Oncoprotein with Many Functions. Mol Cancer 2011, 10, 140–140, doi:10.1186/1476-4598-10-140.
  2. Rosenberger, S.; Arce, J.D.-C.; Langbein, L.; Steenbergen, R.D.M.; Rösl, F. Alternative Splicing of Human Papillomavirus Type-16 E6/E6* Early MRNA Is Coupled to EGF Signaling via Erk1/2 Activation. Proc National Acad Sci 2010, 107, 7006–7011, doi:10.1073/pnas.1002620107.
  3. Schwartz, S. Papillomavirus Transcripts and Posttranscriptional Regulation. Virology 2013, 445, 187–196, doi:10.1016/j.virol.2013.04.034.

Line 211:  "Multiple additional mechanisms help HPV avoid the immune system" (If you want to use "aid" you need to use the infinitive of avoid – "to avoid")

Thank you for pointing out some English nuances that these native speakers did not even recognize. We have fixed the typographical error and have modified the syntax, as suggested.

Lines 211-218:  The following two sentences virtually negate each other's importance: 

1)  Infection of the basal layer inherently ensures multiple layers of epithelial cells protecting virally infected cells from circulating immune cells

2)  The viral lifecycle has adapted to produce the majority viral proteins once in the apical layers of epithelium, which don’t have as many active immune cells.

The conclusion seems to be that no layers are completely safe from immune cells.  Please revise and strengthen your arguments, or remove.

Thank you for this insightful comment! We want to make sure that our text is clear throughout. The first sentence refers to the basal layer of the epidermis, and the second sentence refers to the most superficial layer of the epidermis. In the intermediate layers of epidermis, virally-infected cells are most susceptible to detection by immune cells. It is during this phase of the cell cycle that viral proteins must minimize their presence to avoid immune-mediated cell death. The text has been revised to better clarify this, in the revised manuscript as follows:

Multiple additional mechanisms help HPV avoid the immune system[62]. Infection of the basal layer inherently ensures that multiple layers of epithelial cells protect virally-infected cells from circulating immune cells (where most immune cells reside)[63]. Similarly, the viral lifecycle has adapted to produce the majority viral proteins once in the apical layers of epithelium, which don’t have as many active immune cells. In the intermediate layer, the non-lytic replicative cycle produces fewer free viral markers that would otherwise alert the innate immune system[64].

  1. McBride, A.A. Human Papillomaviruses: Diversity, Infection and Host Interactions. Nat Rev Microbiol 2022, 20, 95–108, doi:10.1038/s41579-021-00617-5.
  2. Zheng, K.; Egawa, N.; Shiraz, A.; Katakuse, M.; Okamura, M.; Griffin, H.M.; Doorbar, J. The Reservoir of Persistent Human Papillomavirus Infection; Strategies for Elimination Using Anti-Viral Therapies. Viruses 2022, 14, 214, doi:10.3390/v14020214.
  3. Doorbar, J.; Zheng, K.; Aiyenuro, A.; Yin, W.; Walker, C.M.; Chen, Y.; Egawa, N.; Griffin, H.M. Principles of Epithelial Homeostasis Control during Persistent Human Papillomavirus Infection and Its Deregulation at the Cervical Transformation Zone. Curr Opin Virol 2021, 51, 96–105, doi:10.1016/j.coviro.2021.09.014.

Line 240:  "may confound this theory" – please clarify

Thank you for pointing out this ambiguity. We have now included the following clause to clarify our statement:

Patient age, stage at diagnosis, and other factors may confound this theory as the younger demographic afflicted with HPV-related OPSCC may tolerate treatment better than older patients—thus producing better outcomes.

Line 241:  "acceptions" – should be "exceptions"

Thanks again for noting the typographical error. We have now corrected this mistake.

Line 243: "as it does in other HPV related cancers" – please provide a reference

We have added several references to support this fact:

  1. Hausen, H. zur Papillomaviruses and Cancer: From Basic Studies to Clinical Application. Nat Rev Cancer 2002, 2, 342–350, doi:10.1038/nrc798.
  2. Castle, P.E.; Einstein, M.H.; Sahasrabuddhe, V.V. Cervical Cancer Prevention and Control in Women Living with Human Immunodeficiency Virus. Ca Cancer J Clin 2021, 71, 505–526, doi:10.3322/caac.21696.
  3. Wang, C.-C.J.; Palefsky, J.M. HIV/AIDS-Associated Viral Oncogenesis. Canc Treat 2018, 177, 183–209, doi:10.1007/978-3-030-03502-0_7.
  4. Beachler, D.C.; D’Souza, G. Oral Human Papillomavirus Infection and Head and Neck Cancers in HIV-Infected Individuals. Curr Opin Oncol 2013, 25, 503–510, doi:10.1097/cco.0b013e32836242b4.

Lines 244-246:  "Additionally, exposure to other risk factors in HPV-positive patients, such as chronic tobacco use, nullifies this benefit. This is thought to be due to baseline chromosomal instability in patients exposed to tobacco products than those who are not." – please provide a reference.  (Reference 51 does not address tobacco and chromosomal instability.)

Thank you! We have added several references to support this fact:

  1. Chen, S.Y.; Massa, S.; Mazul, A.L.; Kallogjeri, D.; Yaeger, L.; Jackson, R.S.; Zevallos, J.; Pipkorn, P. The Association of Smoking and Outcomes in HPV-Positive Oropharyngeal Cancer: A Systematic Review. Am J Otolaryng 2020, 41, 102592, doi:10.1016/j.amjoto.2020.102592.
  2. Villepelet, A.; Hugonin, S.; Atallah, S.; Job, B.; Baujat, B.; Guily, J.L.St.; Lacave, R. Effects of Tobacco Abuse on Major Chromosomal Instability in Human Papilloma Virus 16-Positive Oropharyngeal Squamous Cell Carcinoma. Int J Oncol 2019, 55, 527–535, doi:10.3892/ijo.2019.4826.
  3. Lechner, M.; Liu, J.; Masterson, L.; Fenton, T.R. HPV-Associated Oropharyngeal Cancer: Epidemiology, Molecular Biology and Clinical Management. Nat Rev Clin Oncol 2022, 1–22, doi:10.1038/s41571-022-00603-7.

Lines 257-258:  "In cervicovaginal studies, increased species richness of the vaginal flora were linked to higher prevalence of HPV infection" – please provide a reference

Thank you! We have added several references to support this fact:

  1. Mitra, A.; MacIntyre, D.A.; Marchesi, J.R.; Lee, Y.S.; Bennett, P.R.; Kyrgiou, M. The Vaginal Microbiota, Human Papillomavirus Infection and Cervical Intraepithelial Neoplasia: What Do We Know and Where Are We Going Next? Microbiome 2016, 4, 58, doi:10.1186/s40168-016-0203-0.
  2. Zhang, Y.; D’Souza, G.; Fakhry, C.; Bigelow, E.O.; Usyk, M.; Burk, R.D.; Zhao, N. Oral HPV Associated with Differences in Oral Microbiota Beta Diversity and Microbiota Abundance. J Infect Dis 2022, doi:10.1093/infdis/jiac010.

Lines 269-270:  "Tobacco use may facilitate a similar environment, explaining why cigarette smoking is a negative prognostic factor, even in HPV positive cancers." – please provide a reference.  If there is no evidence that tobacco facilitates a similar environment, please remove.

Thank you for the critical feedback. The statement above is a conjecture based on our research. We have added a clause to clarify that this is a hypothesis rather than fact. The sentence now reads:

We suspect that tobacco use may facilitate a similar environment, explaining why cigarette smoking is a negative prognostic factor, even in HPV positive cancers[104,105].

  1. Bouland, C.; Dequanter, D.; Lechien, J.R.; Hanssens, C.; Aubain, N.D.S.; Digonnet, A.; Javadian, R.; Yanni, A.; Rodriguez, A.; Loeb, I.; et al. Prognostic Significance of a Scoring System Combining P16, Smoking, and Drinking Status in a Series of 131 Patients with Oropharyngeal Cancers. Int J Otolaryngology 2021, 2021, 8020826, doi:10.1155/2021/8020826.
  2. Ference, R.; Liao, D.; Gao, Q.; Mehta, V. Impact of Smoking on Survival Outcomes in HPV-Related Oropharyngeal Carcinoma: A Meta-Analysis. Otolaryngology Head Neck Surg 2020, 163, 1114–1122, doi:10.1177/0194599820931803.

Lines 278-289:  Please provide references for the following:  "Epstein-Barr virus (EBV) is a double stranded DNA virus in the herpesvirus family. It was discovered by Dr. Epstein and his assistant, Ms. Barr, during a global investigation into the cause of Burkitt’s lymphoma in the 1960s. Since then, in addition to Burkitt’s lymphoma, EBV has been definitively linked to several other cancers, including gastric carcinoma, t-cell lymphoma, and nasopharyngeal carcinoma (NPC). NPC is a subtype of squamous cell carcinoma born from the epithelium of the nasopharynx (the tubular passage posterior to the nasal cavity that connects inferiorly to the oropharynx). The most common site for NPC is the fossa of Rosenmüller, the pharyngeal recess located behind the torus tubarius (the medial cartilaginous end of the Eustachian tube). Given this location, symptoms of early-stage NPC seem non-specific and include congestion, headache, aural fullness, and/or mild epistaxis. Due to the vague qualities of these symptoms, NPC is often diagnosed at later stages. Hence, the most common presentation of NPC is a painless neck mass—the result of regional cervical lymphatic spread." The above lack of references was egregious, and showed that unsupported statements were a major problem in this manuscript.  I did not continue to list individual instances of this problem.

Thank you! As noted above and throughout our response, we have now bolstered our text with numerous references in the revised manuscript. In this case, the text now reads as:

Epstein-Barr virus (EBV) is a double stranded DNA virus in the herpesvirus family[109]. It was discovered by Dr. Epstein and his assistant, Ms. Barr, during a global investigation into the cause of Burkitt’s lymphoma in the 1960s[110]. Since then, in addition to Burkitt’s lymphoma, EBV has been definitively linked to several other cancers, including gastric carcinoma, t-cell lymphoma, and nasopharyngeal carcinoma (NPC)[111]. NPC is a subtype of squamous cell carcinoma born from the epithelium of the nasopharynx (the tubular passage posterior to the nasal cavity that connects inferiorly to the oropharynx)[112]. The most common site for NPC is the fossa of Rosenmüller, the pharyngeal recess located behind the torus tubarius (the medial cartilaginous end of the Eustachian tube). Given this location, symptoms of early-stage NPC seem non-specific and include congestion, headache, aural fullness, and/or mild epistaxis[113]. Due to the vague qualities of these symptoms, NPC is often diagnosed at later stages. Hence, the most common presentation of NPC is a painless neck mass—the result of regional cervical lymphatic spread[114]. Following careful workup and staging, treatment typically consists of radiotherapy with or without chemotherapy[115]. Although patients with early-stage disease have good outcomes with radiotherapy alone (87–93% 5-year survival), more intensive treatment strategies combining radiotherapy with chemotherapy are required to manage later-stage disease (63–81% 5-year survival)[37].

  1. Bauer, M.; Jasinski-Bergner, S.; Mandelboim, O.; Wickenhauser, C.; Seliger, B. Epstein–Barr Virus—Associated Malignancies and Immune Escape: The Role of the Tumor Microenvironment and Tumor Cell Evasion Strategies. Cancers 2021, 13, 5189, doi:10.3390/cancers13205189.
  2. Epstein, M.A.; Achong, B.G.; Barr, Y.M. Virus Particles In Cultured Lymphoblasts from Burkitt’s Lymphoma. Lancet 1964, 283, 702–703, doi:10.1016/s0140-6736(64)91524-7.
  3. Hsu, J.L.; Glaser, S.L. Epstein–Barr Virus-Associated Malignancies: Epidemiologic Patterns and Etiologic Implications. Crit Rev Oncol Hemat 2000, 34, 27–53, doi:10.1016/s1040-8428(00)00046-9.
  4. Chen, Y.-P.; Chan, A.T.C.; Le, Q.-T.; Blanchard, P.; Sun, Y.; Ma, J. Nasopharyngeal Carcinoma. Lancet 2019, 394, 64–80, doi:10.1016/s0140-6736(19)30956-0.
  5. Chua, M.L.K.; Wee, J.T.S.; Hui, E.P.; Chan, A.T.C. Nasopharyngeal Carcinoma. Lancet 2016, 387, 1012–1024, doi:10.1016/s0140-6736(15)00055-0.
  6. (Stan), D.J.; Niculet, E.; Lungu, M.; Onisor, C.; Rebegea, L.; Vesa, D.; Bezman, L.; Bujoreanu, F.C.; Sarbu, M.I.; Mihailov, R.; et al. Nasopharyngeal Carcinoma: A New Synthesis of Literature Data (Review). Exp Ther Med 2022, 23, 136, doi:10.3892/etm.2021.11059.
  7. Lee, A.W.M.; Ng, W.T.; Chan, L.L.K.; Hung, W.M.; Chan, C.C.C.; Sze, H.C.K.; Chan, O.S.H.; Chang, A.T.Y.; Yeung, R.M.W. Evolution of Treatment for Nasopharyngeal Cancer – Success and Setback in the Intensity-Modulated Radiotherapy Era. Radiother Oncol 2014, 110, 377–384, doi:10.1016/j.radonc.2014.02.003.
  8. NCCN NCCN Guidelines Version 1.2022 Head and Neck Cancers 2022.

Reviewer 2 Report

 The manuscript is a review of the association of HPV with OPSCC and EBV with NPC. They thoroughly review the literature and highlight how perturbations of the pharyngeal microbiome may impact host-microbiome-tumor-viral interactions—leading to tumor growth.

  The manuscript is interesting and informative. The manuscript is acceptable in its present form.

Author Response

Reviewer 2

Comments and Suggestions for Authors

The manuscript is a review of the association of HPV with OPSCC and EBV with NPC. They thoroughly review the literature and highlight how perturbations of the pharyngeal microbiome may impact host-microbiome-tumor-viral interactions—leading to tumor growth.

The manuscript is interesting and informative. The manuscript is acceptable in its present form.

Thank you so much for taking the time to read and review our manuscript. We are excited and humbled to contribute these efforts to the existing literature, and we extend our sincerest gratitude for your support.

Reviewer 3 Report

I had a great pleasure reading and reviewing the paper entitled: “It Takes Two to Tango; a Review of Oncogenic Virus and Host Microbiome Associated Inflammation in Head and Neck Cancer”. The topic is of great importance, and the presentation of the paper is impressive. The authors did a great effort in putting this paper together. However, for a review paper, the accuracy of the reported results as well as citing the proper source is critical, which certain deficiencies in this paper on doing so. Highlighted below are a few of these to make the authors aware of what is needed to improve the paper and its accuracy.

  • Page 2, lines 63-64: “100% of nasopharyngeal cancers are linked to EBV”. Are the authors referring to only non-keratinizing nasopharyngeal carcinomas?
  • Page 2, lines 65-67: “It is estimated 90% of adults have been exposed to HPV16; seventy percent of adult have evidence of prior infection as demonstrated by the presence of HPV16 anti-bodies in their blood”. This does not appear to be supported by the citation, please find a proper citation/original paper or restate what is mentioned in the cited paper. For example, the paper states that "the majority of oral cancers (approximately 90%) caused by HPV are identified as HPV 16 positive".
  • Page 4, lines 122-123: “specifically, about 70% of these cancers are caused by HPV16 and HPV18 genotypes”. This is a correct statement, but the two cited articles do not support this statement.
  • Page 4, lines 134-136: “However, the two diseases are distinct in that HPV is implicated in about 70% of OPSCC versus approximately 10% of OCSCC.” Again, the numbers reported here, are not supported by the referenced article.
  • Page 5, lines 149-150: “active oral HPV infection have found that more than 3% of adult men and 1% of adult woman have HPV16 detectable in their saliva at any one time”. Again, no evidence from the paper to support this statement.

Other issues:

  • In the methods, the authors stated that they used the MeSH terms, however few terms were not MeSH terms (specifically EBV and HPV).
  • It was hard to replicate the 248 included because the author didn’t state the and/or criteria they used. It will be important to state these, at least as a supplement, so others can replicate the exact same results.
  • It would be helpful to have a paragraph with proper citations to explain figure 1, specifically showing with evidence the microorganism for the Eubiosis.
  • Minor issue: On page 2, line 50, the name of the bacteria is Porphyromonas and not Porphyromas
  • Minor issue: The sentence on page 2, lines 59-62 “Two pertinent examples ….”, needs a proper citation.
  • Recommendation: To include other sources than PubMed like Embase. This is only a recommendation, as the authors mentioned they added few articles from Google Scholar and Web of Science, but these were not added in a systematic manner.
  • Recommendation: Using a reporting standard like PRIMSA can improve the reporting approach of this paper so the readers can have a full picture of what was done at each step.
  • Recommendation: It is recommended to use the original article that demonstrated the results, rather than citing a review paper that included the results.

Author Response

Reviewer 3

Comments and Suggestions for Authors

I had a great pleasure reading and reviewing the paper entitled: “It Takes Two to Tango; a Review of Oncogenic Virus and Host Microbiome Associated Inflammation in Head and Neck Cancer”. The topic is of great importance, and the presentation of the paper is impressive. The authors did a great effort in putting this paper together. However, for a review paper, the accuracy of the reported results as well as citing the proper source is critical, which certain deficiencies in this paper on doing so. Highlighted below are a few of these to make the authors aware of what is needed to improve the paper and its accuracy.

Thank you for your time and thoughtfulness in reviewing our manuscript. We agree that proper citation is critical to ensuring the quality of this work, and we have made significant improvements throughout the manuscript in this regard.

Specific comments:

Page 2, lines 63-64: “100% of nasopharyngeal cancers are linked to EBV”. Are the authors referring to only non-keratinizing nasopharyngeal carcinomas?

We thank the author for bringing up this subtle but significant point. We were hoping to steer away from the nuances of keratinizing and non-keratinizing malignancies (in both the oro– and naso– pharynx); however, this point bears clarification. Indeed, nearly 100% of patients with non-keratinizing nasopharyngeal carcinomas (NPC) are EBV-positive; however, this number is closer to 90% for keratinizing NPC. We have changed the text and added additional citations, as shown below, to clarify this point:

Likewise, nearly all patients with nasopharyngeal carcinoma (NPC) have been exposed to EBV, and EBV transcripts or positive serology is detected in about ninety percent of such patients (more frequently in the non-keritinizing, as opposed to keratinizing, subtype of NPC)[12–14].

  1. Goldenberg, D.; Golz, A.; Netzer, A.; Rosenblatt, E.; Rachmiel, A.; Goldenberg, R.F.; Joachims, H.Z. Epstein-Barr Virus and Cancers of the Head and Neck. Am J Otolaryng 2001, 22, 197–205, doi:10.1053/ajot.2001.23429.
  2. Tsang, C.M.; Deng, W.; Yip, Y.L.; Zeng, M.-S.; Lo, K.W.; Tsao, S.W. Epstein-Barr Virus Infection and Persistence in Nasopharyngeal Epithelial Cells. Chin J Cancer 2014, 33, 549–555, doi:10.5732/cjc.014.10169.
  3. Huang, W.B.; Chan, J.Y.W.; Liu, D.L. Human Papillomavirus and World Health Organization Type III Nasopharyngeal Carcinoma: Multicenter Study from an Endemic Area in Southern China. Cancer 2018, 124, 530–536, doi:10.1002/cncr.31031.

Page 2, lines 65-67: “It is estimated 90% of adults have been exposed to HPV16; seventy percent of adult have evidence of prior infection as demonstrated by the presence of HPV16 anti-bodies in their blood”. This does not appear to be supported by the citation, please find a proper citation/original paper or restate what is mentioned in the cited paper. For example, the paper states that "the majority of oral cancers (approximately 90%) caused by HPV are identified as HPV 16 positive".

Thank you for noting this discrepancy. We have modified the text to reflect the findings of Dr. Gillison, who established the link between HPV and OPSCC. Additionally, we have qualified the statement, acknowledging that exposure rates are likely much higher than those of active infections. The text has been amended as such:

It is estimated that about 15% of adults have a high-risk HPV oropharyngeal infection in their lifetime, with many more likely having some type of exposure given infection rates at other anatomic sites[11,16,17].

  1. Gillison, M.L. Human Papillomavirus-Related Diseases: Oropharynx Cancers and Potential Implications for Adolescent HPV Vaccination. J Adolescent Health 2008, 43, S52–S60, doi:10.1016/j.jadohealth.2008.07.002.
  2. Hausen, H. zur Papillomaviruses and Cancer: From Basic Studies to Clinical Application. Nat Rev Cancer 2002, 2, 342–350, doi:10.1038/nrc798.
  3. Han, J.J.; Beltran, T.H.; Song, J.W.; Klaric, J.; Choi, Y.S. Prevalence of Genital Human Papillomavirus Infection and Human Papillomavirus Vaccination Rates Among US Adult Men: National Health and Nutrition Examination Survey (NHANES) 2013-2014. Jama Oncol 2017, 3, 810, doi:10.1001/jamaoncol.2016.6192.

Page 4, lines 122-123: “specifically, about 70% of these cancers are caused by HPV16 and HPV18 genotypes”. This is a correct statement, but the two cited articles do not support this statement.

Thank you for the keen eye! The references have been updated in the revised manuscript:

Specifically, about 70% of these cancers are caused by HPV16 and HPV18 genotypes[10,11,30,31]; these strains are detected significantly more often in biopsies of malignant tonsillar disease than in benign tonsillar pathology[32,33].

  1. Marur, S.; D’Souza, G.; Westra, W.H.; Forastiere, A.A. HPV-Associated Head and Neck Cancer: A Virus-Related Cancer Epidemic. Lancet Oncol 2010, 11, 781–789, doi:10.1016/s1470-2045(10)70017-6.
  2. Gillison, M.L. Human Papillomavirus-Related Diseases: Oropharynx Cancers and Potential Implications for Adolescent HPV Vaccination. J Adolescent Health 2008, 43, S52–S60, doi:10.1016/j.jadohealth.2008.07.002.
  3. Santos, J.M.O.; Silva, S.P. da; Costa, N.R.; Costa, R.M.G. da; Medeiros, R. The Role of MicroRNAs in the Metastatic Process of High-Risk HPV-Induced Cancers. Cancers 2018, 10, 493, doi:10.3390/cancers10120493.
  4. Gupta, S.; Kumar, P.; Das, B.C. HPV: Molecular Pathways and Targets. Curr Prob Cancer 2018, 42, 161–174, doi:10.1016/j.currproblcancer.2018.03.003.
  5. Ilmarinen, T.; Munne, P.; Hagström, J.; Haglund, C.; Auvinen, E.; Virtanen, E.I.; Haesevoets, A.; Speel, E.J.M.; Aaltonen, L.-M. Prevalence of High-Risk Human Papillomavirus Infection and Cancer Gene Mutations in Nonmalignant Tonsils. Oral Oncol 2017, 73, 77–82, doi:10.1016/j.oraloncology.2017.08.010.
  6. Herberhold, S.; Hellmich, M.; Panning, M.; Bartok, E.; Silling, S.; Akgül, B.; Wieland, U. Human Polyomavirus and Human Papillomavirus Prevalence and Viral Load in Non-Malignant Tonsillar Tissue and Tonsillar Carcinoma. Med Microbiol Immun 2016, 206, 93–103, doi:10.1007/s00430-016-0486-6.

Page 4, lines 134-136: “However, the two diseases are distinct in that HPV is implicated in about 70% of OPSCC versus approximately 10% of OCSCC.” Again, the numbers reported here, are not supported by the referenced article.

In the cited article, it is stated that: “…in North America and Europe, transcriptionally active, high-risk HPV (as evidenced by either quantitative reverse-transcriptase polymerase chain reaction or in situ hybridization-based methods for high-risk HPV E6 and E7 mRNA) has been detected in only about 0% to 9% of OC-SCC cases examined.” The same article references a meta-analysis finding: “the HPV-attributable fraction is approximately 40% for OP-SCC and 7% to 16% for OC-SCC” but notes that in more recent studies that “HPV has been estimated to account for… >60% of cases.” However, given the reviewer’s concerns, we have softened the language and added more supporting citations:

However, the two diseases are distinct in that HPV is implicated in a majority of new OPSCC cases versus only 7-14% of OCSCC[10,11,34].

  1. Marur, S.; D’Souza, G.; Westra, W.H.; Forastiere, A.A. HPV-Associated Head and Neck Cancer: A Virus-Related Cancer Epidemic. Lancet Oncol 2010, 11, 781–789, doi:10.1016/s1470-2045(10)70017-6.
  2. Gillison, M.L. Human Papillomavirus-Related Diseases: Oropharynx Cancers and Potential Implications for Adolescent HPV Vaccination. J Adolescent Health 2008, 43, S52–S60, doi:10.1016/j.jadohealth.2008.07.002.
  3. Chi, A.C.; Day, T.A.; Neville, B.W. Oral Cavity and Oropharyngeal Squamous Cell Carcinoma—an Update. Ca Cancer J Clin 2015, 65, 401–421, doi:10.3322/caac.21293.

Page 5, lines 149-150: “active oral HPV infection have found that more than 3% of adult men and 1% of adult woman have HPV16 detectable in their saliva at any one time”. Again, no evidence from the paper to support this statement.

The appropriate citations have been included in the revised manuscript:

Studies evaluating the prevalence of active oral HPV infection have found that 1-5% of adults have HPV16 detectable in their saliva at any one time[41,42].

  1. D’Souza, G.; Kreimer, A.R.; Viscidi, R.; Pawlita, M.; Fakhry, C.; Koch, W.M.; Westra, W.H.; Gillison, M.L. Case–Control Study of Human Papillomavirus and Oropharyngeal Cancer. New Engl J Medicine 2007, 356, 1944–1956, doi:10.1056/nejmoa065497.
  2. Kreimer, A.R.; Bhatia, R.K.; Messeguer, A.L.; González, P.; Herrero, R.; Giuliano, A.R. Oral Human Papillomavirus in Healthy Individuals; A Systematic Review of the Literature. Sex Transm Dis 2010, 37, 386–391, doi:10.1097/olq.0b013e3181c94a3b.

It would be helpful to have a paragraph with proper citations to explain figure 1, specifically showing with evidence the microorganism for the Eubiosis.

Citations were added to the legend for Figure 1, as suggested by yourself and Reviewer 1. These are copied below:

Figure 1: overview of environment-host-microbiome-tumor interactions

Schematic overview of the interplay between the environment, the oral microbiome, human papillomavirus (HPV), a tonsillar tumor, and host inflammation. Environmental factors, such as tobacco smoke, alter the oral microbiome[156–158]. Meanwhile, bacterial shifts in the oral microbiome modulate viral proliferation and infection[159]. The interplay of these microbiological factors, a tumor, and the host immune system is complex[160–162]. Figure generated using BioRender.

  1. Colman, G.; Beighton, D.; Chalk, A.J.; Wake, S. Cigarette Smoking and the Microbial Flora of the Mouth*. Aust Dent J 1976, 21, 111–118, doi:10.1111/j.1834-7819.1976.tb02833.x.
  2. Morris, A.; Beck, J.M.; Schloss, P.D.; Campbell, T.B.; Crothers, K.; Curtis, J.L.; Flores, S.C.; Fontenot, A.P.; Ghedin, E.; Huang, L.; et al. Comparison of the Respiratory Microbiome in Healthy Nonsmokers and Smokers. Am J Resp Crit Care 2013, 187, 1067–1075, doi:10.1164/rccm.201210-1913oc.
  3. Mason, M.R.; Preshaw, P.M.; Nagaraja, H.N.; Dabdoub, S.M.; Rahman, A.; Kumar, P.S. The Subgingival Microbiome of Clinically Healthy Current and Never Smokers. Isme J 2015, 9, 268–272, doi:10.1038/ismej.2014.114.
  4. Tada, A.; Senpuku, H. The Impact of Oral Health on Respiratory Viral Infection. Dent J 2021, 9, 43, doi:10.3390/dj9040043.
  5. Shetty, S.S.; Padam, K.S.R.; Hunter, K.D.; Kudva, A.; Radhakrishnan, R. Biological Implications of the Immune Factors in the Tumour Microenvironment of Oral Cancer. Arch Oral Biol 2021, 133, 105294, doi:10.1016/j.archoralbio.2021.105294.
  6. Zong, Y.; Zhou, Y.; Liao, B.; Liao, M.; Shi, Y.; Wei, Y.; Huang, Y.; Zhou, X.; Cheng, L.; Ren, B. The Interaction Between the Microbiome and Tumors. Front Cell Infect Mi 2021, 11, 673724, doi:10.3389/fcimb.2021.673724.
  7. Kurago, Z.; Loveless, J. Microbial Colonization and Inflammation as Potential Contributors to the Lack of Therapeutic Success in Oral Squamous Cell Carcinoma. Frontiers Oral Heal 2021, 2, 739499, doi:10.3389/froh.2021.739499.

Minor issue: On page 2, line 50, the name of the bacteria is Porphyromonas and not Porphyromas

Thank you for noting this typo; the misspelling has been corrected.

Minor issue: The sentence on page 2, lines 59-62 “Two pertinent examples ….”, needs a proper citation.

Thank you! In the revised manuscript we have included citations following this sentence and throughout the manuscript following statements of fact. The citations in this case were:

Two pertinent examples are the cancer-related viruses, human papillomavirus (HPV) and Epstein–Barr virus (EBV), which colonize the pharynx and are responsible for a growing proportion of head and neck cancers[8].

  1. Chaturvedi, A.K.; Engels, E.A.; Pfeiffer, R.M.; Hernandez, B.Y.; Xiao, W.; Kim, E.; Jiang, B.; Goodman, M.T.; Sibug-Saber, M.; Cozen, W.; et al. Human Papillomavirus and Rising Oropharyngeal Cancer Incidence in the United States. J Clin Oncol 2011, 29, 4294–4301, doi:10.1200/jco.2011.36.4596.

Recommendation: It is recommended to use the original article that demonstrated the results, rather than citing a review paper that included the results.

Thank you for this suggestion. In adding citations, we have now tried to focus on primary, as opposed to secondary (review) literature.

In the methods, the authors stated that they used the MeSH terms, however few terms were not MeSH terms (specifically EBV and HPV).

See below.

It was hard to replicate the 248 included because the author didn’t state the and/or criteria they used. It will be important to state these, at least as a supplement, so others can replicate the exact same results.

See below.

Recommendation: To include other sources than PubMed like Embase. This is only a recommendation, as the authors mentioned they added few articles from Google Scholar and Web of Science, but these were not added in a systematic manner.

See below.

Recommendation: Using a reporting standard like PRIMSA can improve the reporting approach of this paper so the readers can have a full picture of what was done at each step.

We have grouped these excellent criticisms as they are related and best can be addressed together. As noted in our response to Reviewer 1, the goal of the systematic-style search described in the methods was to identify studies (linking the inflammation, the microbiome, EBV/HPV, and head and neck cancer) that we may not have otherwise found via a more traditional (narrative) style review. We were not aiming to perform a true systematic review; and, had we done so, we indeed would have included 1) more detailed methods, 2) a list of all identified papers, and 3) a PRISMA diagram (as your expertly suggest). We have added more details about the MeSH terms in the Method section (copied below) so that our search can be replicated.

In order to identify studies linking the inflammation, the microbiome, EBV/HPV, and head and neck cancer, we conducted a systematic-style search in PubMed. The terms pharynx, oropharynx, nasopharynx (and related anatomy), viral infections, inflammation, EBV, HPV, and carcinogenesis were queried. The specific search query was: ("Pharynx"[Mesh:NoExp] OR "Oropharynx"[Mesh] OR "Nasopharynx"[Mesh] OR "Palate, Soft"[Mesh] OR "Eustachian Tube"[Mesh]) AND("Viruses"[Mesh] OR "Virology"[Mesh] OR "Virome"[Mesh] OR "Infections"[Mesh] OR "Inflammation"[Mesh] OR "Epstein-Barr Virus Infections"[Mesh] OR "Papillomaviridae"[Mesh] "Epstein Barr"[tiab] OR "EBV infection"[tiab] OR "EBV infections"[tiab] OR papillomaviridae[tiab] OR "human papilloma virus"[tiab] OR "human papilloma viruses"[tiab] OR "human papillomavirus"[tiab] OR HPV[tiab]) AND ("Carcinogenesis"[Mesh] OR "Oropharyngeal Neoplasms"[Mesh] OR "Nasopharyngeal Neoplasms"[Mesh]). No date limits were imposed, and the last search was performed on October 21st, 2021.

Our search initially resulted in 248 records. Titles were then independently evaluated for relevance by authors JG and MGM (using a citation manager), resulting in 114 relevant abstracts. Whole manuscripts were further screened for relevance. Non-original articles, review articles, non-human studies, and abstract-only publications were then excluded. Additionally, studies published in languages other than English were excluded. Interrater conflicts were assessed and re-designated relevant or irrelevant by MGM, leaving 45 articles for review. Any pertinent topic areas that were uncovered following this thorough search were supplemented with specific searches in complementary databases, such as Google Scholar, PubMed, and Web of Science. Additional searches to specifically link the microbiome, HPV, and OPSCC yielded

Round 2

Reviewer 1 Report

This manuscript is much improved since its original submission.  While some additional references, clarifications, and corrections are necessary, it is generally a well written manuscript that will be useful to the field.

Specific comments:

Abstract:  Please explain or correct these numbers presented in the abstract: 

 7% of the population is infected at any time; 1% develop OPSCC.

The American population is roughly 300,000,000.  1% of 7% of that number is roughly 210,000 cases.  However, current yearly OPSCC incidence is less than 10% of that number.  There are various ways to analyze this, but I don’t think any suggest a cancer rate as high as 1% of infected people.

Figures:  They are not labeled and do not have figure legends.  The legends should at least refer to the text for explanation of their contents.

p. 7 "Stimulation of EGFR also activates down steam products" -- should read "downstream"

 p.8 "it also preferentially induces aneuploidy of E7 expressing genes via DNMT1[54]" – reference 54 does not address aneuploidy

 p. 8  The following remains illogical:

"Infection of the basal layer inherently ensures that multiple layers of epithelial cells protect virally-infected cells from circulating immune cells (where most immune cells reside)[63]."

As written, this sentence suggests that the presence of virus in the basal cells helps the virus evade the immune system.  However, it isn’t the presence of virus in that layer, close to circulating immune cells, that protects the virus from immune attack.  Rather, it is the relegation of most viral protein production to the upper layers that helps protect the virus – as stated in the sentences that follow the first.  Please delete the first sentence ("Infection of the basal layer inherently ensures that multiple layers of epithelial cells protect virally-infected cells from circulating immune cells (where most immune cells reside)") or explain why having the virus in the basal layer is advantageous in terms of immune evasion.  The second and third sentences make sense on their own (if you delete "Similarly").

p.8 “Impairment of the innate immune system negatively impacts a functional adaptive immune response.”  Please provide a reference.

p. 10 “creates an environments” – environment should be singular

p. 10 “One study identified microbial shifts that either directly (through virulence factors) or indirectly (through oxidative stress) promoted shifts in gene expression and contributed to carcinogenesis[102]”. This study established a correlation, not cause-and-effect as your summary of it suggests. 

p. 11 “infection . . . was lytic and self-contaminated.”  Do you mean “self-contained”?

p. 11 There are several typos in the last paragraph

p. 13 “help promote cell vision” – cell division?  There are several other typos on p. 13

Author Response

Reviewer #1:

This manuscript is much improved since its original submission. While some additional references, clarifications, and corrections are necessary, it is generally a well written manuscript that will be useful to the field.

Thank you for your contributions; your comments have significantly improved the quality of our work.

Abstract: Please explain or correct these numbers presented in the abstract:

7% of the population is infected at any time; 1% develop OPSCC.

The American population is roughly 300,000,000. 1% of 7% of that number is roughly 210,000 cases. However, current yearly OPSCC incidence is less than 10% of that number. There are various ways to analyze this, but I don’t think any suggest a cancer rate of 1% of infected people.

Thank you for noting that this statement is confusing. The 7% figure refers to all oral HPV infection. The 1% refers to the conversation rate for high-risk HPV. We have revised the statement to focus on high-risk HPV and to clear up this confusion:

While HPV is a ubiquitously present virus (with about 1% of the population having high-risk oral HPV infection at any one time), less than 1% of those infected with high-risk strains develop OPSCC— suggesting that additional cofactors or coinfections may be required.

The relevant citations would be:

  • Sonawane K, Suk R, Chiao EY, et al. Oral Human Papillomavirus Infection: Differences in Prevalence Between Sexes and Concordance With Genital Human Papillomavirus Infection, NHANES 2011 to 2014. Ann Intern Med. 2017;167(10):714. doi:10.7326/m17-1363
  • D’Souza G, Kreimer AR, Viscidi R, et al. Case–Control Study of Human Papillomavirus and Oropharyngeal Cancer. New Engl J Medicine. 2007;356(19):1944-1956. doi:10.1056/nejmoa065497

Figures: They are not labeled and do not have figure legends. The legends should at least refer to the text for explanation of their contents.

We apologize for this confusion and believe that this is the result of an editorial error in typesetting. Our figures are labeled and the legends (copied below) were extensively revised during the last round of revisions.

Figure 1: overview of environment-host-microbiome-tumor interactions

Schematic overview of the interplay between the environment, the oral microbiome, human papillomavirus (HPV), a tonsillar tumor, and host inflammation. Environmental factors, such as tobacco smoke, alter the oral microbiome[7,160–164]. Meanwhile, bacterial shifts in the oral microbiome modulate viral proliferation and infection[165]. The interplay of these microbiological factors, a tumor, and the host immune system is complex[166–168]. Figure generated using BioRender.

Figure 2: interplay of HPV, the microbiome, and the oropharynx

Overview of the HPV lifecycle within a tonsillar tumor. Microabrasions in tonsillar epithelium lead to HPV infection of basilar and epithelial cells. Basilar cells undergo lateral expansion during the non-lytic phase of the viral cycle, primarily mediated by E5[43]—which modulates EGFR signaling[169], downregulates CD1d[45], prevents MHCI trafficking[170], and regulates autophagy[171]. Epithelial cells undergo proliferation and transformation prior to the lytic phase of the virus’s life cycle, primarily mediated by E6 (which degrades p53) and E7 (which similarly degrades Rb)[172]. E6 and E7 also affect cell division via TERT and methylation[173,174]. Figure generated using BioRender.

Figure 3: growth and immune evasion of EBV-mediated NPC

Typical pathway for EBV infection and cellular transformation (top). Mechanisms of EBV-mediated cellular transformation and immune evasion (bottom). EBV alters immune checkpoints, including TIM-3 and TRIM29[175]. The virus also alters the immune system via modulation of NFkB.[148]. EBV-specific products LMP1[147] and LMP2[176] also alter the cell cycle and aid in immune evasion[113]. Figure generated using BioRender.

  1. Kumpitsch, C.; Koskinen, K.; Schöpf, V.; Moissl-Eichinger, C. The Microbiome of the Upper Respiratory Tract in Health and Disease. Bmc Biol 2019, 17, 87, doi:10.1186/s12915-019-0703-z.
  2. Venuti, A.; Paolini, F.; Nasir, L.; Corteggio, A.; Roperto, S.; Campo, M.S.; Borzacchiello, G. Papillomavirus E5: The Smallest Oncoprotein with Many Functions. Mol Cancer 2011, 10, 140–140, doi:10.1186/1476-4598-10-140.
  3. Miura, S.; Kawana, K.; Schust, D.J.; Fujii, T.; Yokoyama, T.; Iwasawa, Y.; Nagamatsu, T.; Adachi, K.; Tomio, A.; Tomio, K.; et al. CD1d, a Sentinel Molecule Bridging Innate and Adaptive Immunity, Is Downregulated by the Human Papillomavirus (HPV) E5 Protein: A Possible Mechanism for Immune Evasion by HPV. J Virol 2010, 84, 11614–11623, doi:10.1128/jvi.01053-10.
  4. Bauer, M.; Jasinski-Bergner, S.; Mandelboim, O.; Wickenhauser, C.; Seliger, B. Epstein–Barr Virus—Associated Malignancies and Immune Escape: The Role of the Tumor Microenvironment and Tumor Cell Evasion Strategies. Cancers 2021, 13, 5189, doi:10.3390/cancers13205189.
  5. Wang, L.; Ning, S. New Look of EBV LMP1 Signaling Landscape. Cancers 2021, 13, 5451, doi:10.3390/cancers13215451.
  6. Senten, J.R. van; Fan, T.S.; Siderius, M.; Smit, M.J. Viral G Protein-Coupled Receptors as Modulators of Cancer Hallmarks. Pharmacol Res 2020, 156, 104804, doi:10.1016/j.phrs.2020.104804.
  7. Colman, G.; Beighton, D.; Chalk, A.J.; Wake, S. Cigarette Smoking and the Microbial Flora of the Mouth*. Aust Dent J 1976, 21, 111–118, doi:10.1111/j.1834-7819.1976.tb02833.x.
  8. Morris, A.; Beck, J.M.; Schloss, P.D.; Campbell, T.B.; Crothers, K.; Curtis, J.L.; Flores, S.C.; Fontenot, A.P.; Ghedin, E.; Huang, L.; et al. Comparison of the Respiratory Microbiome in Healthy Nonsmokers and Smokers. Am J Resp Crit Care 2013, 187, 1067–1075, doi:10.1164/rccm.201210-1913oc.
  9. Mason, M.R.; Preshaw, P.M.; Nagaraja, H.N.; Dabdoub, S.M.; Rahman, A.; Kumar, P.S. The Subgingival Microbiome of Clinically Healthy Current and Never Smokers. Isme J 2015, 9, 268–272, doi:10.1038/ismej.2014.114.
  10. Sampaio-Maia, B.; Caldas, I.M.; Pereira, M.L.; Pérez-Mongiovi, D.; Araujo, R. Chapter Four The Oral Microbiome in Health and Its Implication in Oral and Systemic Diseases. Adv Appl Microbiol 2016, 97, 171–210, doi:10.1016/bs.aambs.2016.08.002.
  11. Lim, Y.; Totsika, M.; Morrison, M.; Punyadeera, C. Oral Microbiome: A New Biomarker Reservoir for Oral and Oropharyngeal Cancers. Theranostics 2017, 7, 4313–4321, doi:10.7150/thno.21804.
  12. Tada, A.; Senpuku, H. The Impact of Oral Health on Respiratory Viral Infection. Dent J 2021, 9, 43, doi:10.3390/dj9040043.
  13. Shetty, S.S.; Padam, K.S.R.; Hunter, K.D.; Kudva, A.; Radhakrishnan, R. Biological Implications of the Immune Factors in the Tumour Microenvironment of Oral Cancer. Arch Oral Biol 2021, 133, 105294, doi:10.1016/j.archoralbio.2021.105294.
  14. Zong, Y.; Zhou, Y.; Liao, B.; Liao, M.; Shi, Y.; Wei, Y.; Huang, Y.; Zhou, X.; Cheng, L.; Ren, B. The Interaction Between the Microbiome and Tumors. Front Cell Infect Mi 2021, 11, 673724, doi:10.3389/fcimb.2021.673724.
  15. Kurago, Z.; Loveless, J. Microbial Colonization and Inflammation as Potential Contributors to the Lack of Therapeutic Success in Oral Squamous Cell Carcinoma. Frontiers Oral Heal 2021, 2, 739499, doi:10.3389/froh.2021.739499.
  16. Crusius, K.; Auvinen, E.; Steuer, B.; Gaissert, H.; Alonso, A. The Human Papillomavirus Type 16 E5-Protein Modulates Ligand-Dependent Activation of the EGF Receptor Family in the Human Epithelial Cell Line HaCaT. Exp Cell Res 1998, 241, 76–83, doi:10.1006/excr.1998.4024.
  17. Marchetti, B.; Ashrafi, G.H.; Tsirimonaki, E.; O’Brien, P.M.; Campo, M.S. The Bovine Papillomavirus Oncoprotein E5 Retains MHC Class I Molecules in the Golgi Apparatus and Prevents Their Transport to the Cell Surface. Oncogene 2002, 21, 7808–7816, doi:10.1038/sj.onc.1205885.
  18. Aranda‐Rivera, A.K.; Cruz‐Gregorio, A.; Briones‐Herrera, A.; Pedraza‐Chaverri, J. Regulation of Autophagy by High‐ and Low‐risk Human Papillomaviruses. Rev Med Virol 2021, 31, e2169, doi:10.1002/rmv.2169.
  19. Hoppe-Seyler, K.; Bossler, F.; Braun, J.A.; Herrmann, A.L.; Hoppe-Seyler, F. The HPV E6/E7 Oncogenes: Key Factors for Viral Carcinogenesis and Therapeutic Targets. Trends Microbiol 2018, 26, 158–168, doi:10.1016/j.tim.2017.07.007.
  20. Doorslaer, K.V.; Burk, R.D. Association between HTERT Activation by HPV E6 Proteins and Oncogenic Risk. Virology 2012, 433, 216–219, doi:10.1016/j.virol.2012.08.006.
  21. Burgers, W.A.; Blanchon, L.; Pradhan, S.; Launoit, Y. de; Kouzarides, T.; Fuks, F. Viral Oncoproteins Target the DNA Methyltransferases. Oncogene 2007, 26, 1650–1655, doi:10.1038/sj.onc.1209950.
  22. Biggi, A.F.B.; Oliveira, D.E. de The Epstein-Barr Virus Hacks Immune Checkpoints: Evidence and Consequences for Lymphoproliferative Disorders and Cancers. Biomol 2022, 12, 397, doi:10.3390/biom12030397.
  23. Cen, O.; Longnecker, R. Epstein Barr Virus Volume 2, One Herpes Virus: Many Diseases. Curr Top Microbiol 2015, 391, 151–180, doi:10.1007/978-3-319-22834-1_5.

  1. 7 “Stimulation of EGFR also activated down stream products” – should read “downstream”

Thank you for noticing this typo. The sentence now reads “downstream”.

  1. 8 “it also preferentially induces aneuploidy of E7 expressing genes via DNMT1[54]” – reference 54 does not address aneuploidy

The referenced paper compares methylation patterns in HPV-positive and HPV-negative cancers. Hypermethylation observed in HPV-positive cancers sometimes impacts promoter regions, effectively silencing genes. However, to use “aneuploidy” in this context is incorrect. Thank you for highlighting this. We have revised the sentence to state:

…oncoprotein (E6 and E7) expression has been associated with patterns of hypermethylation which promote E7 expressing genes, often via DNMT1s and maintained via miRNA dysregulation[55].

  1. Camuzi, D.; Simão, T. de A.; Dias, F.; Pinto, L.F.R.; Soares-Lima, S.C. Head and Neck Cancers Are Not Alike When Tarred with the Same Brush: An Epigenetic Perspective from the Cancerization Field to Prognosis. Cancers 2021, 13, 5630, doi:10.3390/cancers13225630.

  1. 8 The following remains illogical:

“Infection of the basal layer… (where most immune cells reside)[63].”

As written, this sentence suggests that the presence of virus in the basal cells helps the virus evade the immune system. However, it isn’t the presence of virus in that layer, close to circulation immune cells, that protects the virus from immune attack. Rather, it is the relegation of most viral protein production to the upper layers that helps protect the virus – as stated in the sentences that follow the first. Please delete the first sentence (“Infection…reside”) or explain why having the virus in the basal layer is advantageous in terms of immune evasion. The second and third sentences make sense on their own (if you delete “Similarly”).

As they were written, these two sentences were contradictory. The incorrect sentence has been deleted. The section now reads:

Infection of the basal layer establishes a reservoir of infected cells. The viral lifecycle has adapted to produce the majority viral proteins once in the apical layers of epithelium, which don’t have as many active immune cells[64].

  1. Zheng, K.; Egawa, N.; Shiraz, A.; Katakuse, M.; Okamura, M.; Griffin, H.M.; Doorbar, J. The Reservoir of Persistent Human Papillomavirus Infection; Strategies for Elimination Using Anti-Viral Therapies. Viruses 2022, 14, 214, doi:10.3390/v14020214.

p.8 Impairment of the innate immune system negatively impacts a functional adaptive immune response.” Please provide a reference.

We have added a citation ([68]) and re-phrased this sentence:

HPV-positive OPSCC have lower expression of TLR 9 and TLR 5 than HPV-negative OPSCC[67], which is important as impairment of the innate immune system negatively impacts a functional adaptive immune response[68].

  1. Jouhi, L.; Datta, N.; Renkonen, S.; Atula, T.; Mäkitie, A.; Haglund, C.; Ahmed, A.; Syrjänen, S.; Grénman, R.; Auvinen, E.; et al. Expression of Toll-like Receptors in HPV-Positive and HPV-Negative Oropharyngeal Squamous Cell Carcinoma--an in Vivo and in Vitro Study. Tumour Biology J Int Soc Oncodevelopmental Biology Medicine 2015, 36, 7755–7764, doi:10.1007/s13277-015-3494-z.
  2. Iwasaki, A.; Medzhitov, R. Toll-like Receptor Control of the Adaptive Immune Responses. Nat Immunol 2004, 5, 987–995, doi:10.1038/ni1112.

p.10 “creates an environments” – environment should be singular

Environment is now singular.

p.10 “one study identified…[102]”. This study established a correlation, not cause-and-effect as your summary of it suggests.

Thank you for highlighting this. We have modified the sentence to say:

One study identified microbial shifts correlated to shifts in gene expression that may contribute to carcinogenesis either directly (through virulence factors) or indirectly (through oxidative stress)[106].

We feel this better represents the correlative relationship described in the cited study.

p.11 “infections… was lytic and self-contaminated”. Do you mean “self-contained”?

This sentence was trying to articulate that scientists previously believed that epithelial cells were not susceptible to chronic infection. Neither “self-contaminated” nor “self-contained” clearly communicate this sentiment. Thus, we have modified the sentence to read:

“It was previously believed epithelial EBV infection… was lytic and time limited.”

p.11 There are several typos in the last paragraph

We have made several changes to this paragraph:

Additional risk factors for NPC include regular consumption of salted fish and Southeast Asian heritage. NPC develops significantly more frequently in patients living in southern China than in other regions of the world—perhaps due to certain human leukocyte antigen (HLA) profiles[121]. Genetic deletions on chromosome three, which impairs cells innate tumor suppression, may also predispose to NPC[124]. Taken together, this supports a genetic predisposition to NPC. However, historical migration patterns and modern epigenetic studies present a compelling argument that environmental exposures play an equally important role in tumorigenesis[125]. Other molecular events suspected to facilitate latent infection, and thus contribute to carcinogenesis, include: hypermethylation of specific genes (e.g., RASSF1A and BLU), deactivation of p16, deactivation of lactoferrin, and upregulation of cyclin D1 production [13,126,127].

  1. Tsang, C.M.; Deng, W.; Yip, Y.L.; Zeng, M.-S.; Lo, K.W.; Tsao, S.W. Epstein-Barr Virus Infection and Persistence in Nasopharyngeal Epithelial Cells. Chin J Cancer 2014, 33, 549–555, doi:10.5732/cjc.014.10169.
  2. Wong, K.C.W.; Hui, E.P.; Lo, K.-W.; Lam, W.K.J.; Johnson, D.; Li, L.; Tao, Q.; Chan, K.C.A.; To, K.-F.; King, A.D.; et al. Nasopharyngeal Carcinoma: An Evolving Paradigm. Nat Rev Clin Oncol 2021, 18, 679–695, doi:10.1038/s41571-021-00524-x.
  3. Bjornevik, K.; Cortese, M.; Healy, B.C.; Kuhle, J.; Mina, M.J.; Leng, Y.; Elledge, S.J.; Niebuhr, D.W.; Scher, A.I.; Munger, K.L.; et al. Longitudinal Analysis Reveals High Prevalence of Epstein-Barr Virus Associated with Multiple Sclerosis. Science 2022, doi:10.1126/science.abj8222.
  4. Pathmanathan, R.; Prasad, U.; Sadler, R.; Flynn, K.; Raab-Traub, N. Clonal Proliferations of Cells Infected with Epstein-Barr Virus in Preinvasive Lesions Related to Nasopharyngeal Carcinoma. New Engl J Medicine 1995, 333, 693–698, doi:10.1056/nejm199509143331103.
  5. Chen, J.; Fu, L.; Zhang, L.-Y.; Kwong, D.L.; Yan, L.; Guan, X.-Y. Tumor Suppressor Genes on Frequently Deleted Chromosome 3p in Nasopharyngeal Carcinoma. Chin J Cancer 2012, 31, 215–222, doi:10.5732/cjc.011.10364.
  6. Chang, E.T.; Ye, W.; Zeng, Y.-X.; Adami, H.-O. The Evolving Epidemiology of Nasopharyngeal Carcinoma. Cancer Epidemiology Prev Biomarkers 2021, 30, 1035–1047, doi:10.1158/1055-9965.epi-20-1702.
  7. Tsang, C.M.; Yip, Y.L.; Lo, K.W.; Deng, W.; To, K.F.; Hau, P.M.; Lau, V.M.Y.; Takada, K.; Lui, V.W.Y.; Lung, M.L.; et al. Cyclin D1 Overexpression Supports Stable EBV Infection in Nasopharyngeal Epithelial Cells. Proc National Acad Sci 2012, 109, E3473-82, doi:10.1073/pnas.1202637109.
  8. Zheng, Y.; Zhang, W.; Ye, Q.; Zhou, Y.; Xiong, W.; He, W.; Deng, M.; Zhou, M.; Guo, X.; Chen, P.; et al. Inhibition of Epstein-Barr Virus Infection by Lactoferrin. J Innate Immun 2012, 4, 387–398, doi:10.1159/000336178.

p.13 “help promote cell visions” – cell divisions? There are several other typos on page 13

Yes, thank you. The sentence now reads “cell divisions”.

Reviewer 3 Report

The authors of the paper entitled “It Takes Two to Tango; a Review of Oncogenic Virus and Host Microbiome Associated Inflammation in Head and Neck Cancer” made a great effort to address my comments and the comments from the other reviewers.

As I previously mentioned (along with reviewer 1), many of the statements in the first version of the manuscript were not supported or restated incorrectly, which is highly concerning for a review paper. That being said, in this new revised version, I believe the authors did well in minimizing such problems, and only a few minor issues were identified:

-         Figure 1: The bacteria listed under Eubiosis could be updated with the most common bacteria found in the oral cavity or include proper citations that list these (PMID: 29158828; PMID: 27926431 and reference number 7 from the manuscript).

-         Please confirm with the search terms as it yields 291 records in PubMed up to 10/21/2021

-         The following statements need proper citation:

a)     Page 7 – “Additionally, E6 has been shown to activate telomerase and stimulate molecular signaling, such as wnt and other cancer hallmark pathways, in infected cells”

b)    Page 9 – “Patient age, stage at diagnosis, and other factors may confound this theory as the younger demographic afflicted with HPV-related OPSCC may tolerate treatment better than older patients—thus producing better outcomes”

Author Response

Review #3:

The authors of the paper entitled “It Takes Two to Tango: A Review of Oncogenic Virus and Host Microbiome Associated Inflammation in Head and Neck Cancer” made a great effort to address my comments and the comments from the other reviewers.

As I previously mentioned (along with reviewer 1), many of the statements in the first version of the manuscript were not supported or restated incorrectly, which is highly concerning for a review paper. That being said, in this new revised version, I believe the authors did well in minimizing such problems, and only a few minor issues were identified:

Thank you for your contributions; your comments have significantly improved the quality of our work.

Figure 1: the bacteria listed under Eubiosis could be updated with the most common bacteria found in the oral cavity or include proper citations that list these (PMID 29158828; PMID: 27926431 and reference number 7 from the manuscript)

The figure legends have been updated accordingly:

Figure 1: overview of environment-host-microbiome-tumor interactions

Schematic overview of the interplay between the environment, the oral microbiome, human papillomavirus (HPV), a tonsillar tumor, and host inflammation. Environmental factors, such as tobacco smoke, alter the oral microbiome[7,160–164]. Meanwhile, bacterial shifts in the oral microbiome modulate viral proliferation and infection[165]. The interplay of these microbiological factors, a tumor, and the host immune system is complex[166–168]. Figure generated using BioRender.

  1. Kumpitsch, C.; Koskinen, K.; Schöpf, V.; Moissl-Eichinger, C. The Microbiome of the Upper Respiratory Tract in Health and Disease. Bmc Biol 2019, 17, 87, doi:10.1186/s12915-019-0703-z.
  2. Colman, G.; Beighton, D.; Chalk, A.J.; Wake, S. Cigarette Smoking and the Microbial Flora of the Mouth*. Aust Dent J 1976, 21, 111–118, doi:10.1111/j.1834-7819.1976.tb02833.x.
  3. Morris, A.; Beck, J.M.; Schloss, P.D.; Campbell, T.B.; Crothers, K.; Curtis, J.L.; Flores, S.C.; Fontenot, A.P.; Ghedin, E.; Huang, L.; et al. Comparison of the Respiratory Microbiome in Healthy Nonsmokers and Smokers. Am J Resp Crit Care 2013, 187, 1067–1075, doi:10.1164/rccm.201210-1913oc.
  4. Mason, M.R.; Preshaw, P.M.; Nagaraja, H.N.; Dabdoub, S.M.; Rahman, A.; Kumar, P.S. The Subgingival Microbiome of Clinically Healthy Current and Never Smokers. Isme J 2015, 9, 268–272, doi:10.1038/ismej.2014.114.
  5. Sampaio-Maia, B.; Caldas, I.M.; Pereira, M.L.; Pérez-Mongiovi, D.; Araujo, R. Chapter Four The Oral Microbiome in Health and Its Implication in Oral and Systemic Diseases. Adv Appl Microbiol 2016, 97, 171–210, doi:10.1016/bs.aambs.2016.08.002.
  6. Lim, Y.; Totsika, M.; Morrison, M.; Punyadeera, C. Oral Microbiome: A New Biomarker Reservoir for Oral and Oropharyngeal Cancers. Theranostics 2017, 7, 4313–4321, doi:10.7150/thno.21804.
  7. Tada, A.; Senpuku, H. The Impact of Oral Health on Respiratory Viral Infection. Dent J 2021, 9, 43, doi:10.3390/dj9040043.
  8. Shetty, S.S.; Padam, K.S.R.; Hunter, K.D.; Kudva, A.; Radhakrishnan, R. Biological Implications of the Immune Factors in the Tumour Microenvironment of Oral Cancer. Arch Oral Biol 2021, 133, 105294, doi:10.1016/j.archoralbio.2021.105294.
  9. Zong, Y.; Zhou, Y.; Liao, B.; Liao, M.; Shi, Y.; Wei, Y.; Huang, Y.; Zhou, X.; Cheng, L.; Ren, B. The Interaction Between the Microbiome and Tumors. Front Cell Infect Mi 2021, 11, 673724, doi:10.3389/fcimb.2021.673724.
  10. Kurago, Z.; Loveless, J. Microbial Colonization and Inflammation as Potential Contributors to the Lack of Therapeutic Success in Oral Squamous Cell Carcinoma. Frontiers Oral Heal 2021, 2, 739499, doi:10.3389/froh.2021.739499.

Please confirm with the search terms as it yields 291 records in PubMed up to 10/21/2021

Thank you for the keen eye. Upon re-review of our citation manager, we initially identified 284 records (as opposed to the 248 previously cited). This was a typo and has been corrected. The difference between our 284 articles and the 291 yielded with your search is that we only used complete entries—records that have been processed by PubMed to include MeSH terms, etc. To replicate the complete record search, one would need to check the appropriate box or add ("1970/01/01"[Date - Completion] : "2021/10/21"[Date – Completion]) to the query. We have clarified the text:

No date limits were imposed, only complete entries were considered, and the last search was performed on October 21st, 2021.

The following statements need proper citation:

Page 7- “Additionally, E6 has been shown to activate telomerase and stimulate molecular signaling, such as wnt and other cancer hallmark pathways, in infected cells”

We have added the following citation:

Additionally, E6 has been shown to activate telomerase and stimulate molecular signaling, such as wnt and other cancer hallmark pathways, in infected cells[50].

  1. Lichtig, H.; Gilboa, D.A.; Jackman, A.; Gonen, P.; Levav-Cohen, Y.; Haupt, Y.; Sherman, L. HPV16 E6 Augments Wnt Signaling in an E6AP-Dependent Manner. Virology 2010, 396, 47–58, doi:10.1016/j.virol.2009.10.011.

Page 9- “Patient age, stage at diagnosis, and other factors may confound this theory as the younger demographic afflicted with HPV-related OPSCC may tolerate treatment better than older patients – thus producing better outcomes”

We have added the following citations:

Patient age, stage at diagnosis, and other factors may confound this theory as the younger demographic afflicted with HPV-related OPSCC may tolerate treatment better than older patients—thus producing better outcomes[77,78].

  1. Machtay, M.; Moughan, J.; Trotti, A.; Garden, A.S.; Weber, R.S.; Cooper, J.S.; Forastiere, A.; Ang, K.K. Factors Associated With Severe Late Toxicity After Concurrent Chemoradiation for Locally Advanced Head and Neck Cancer: An RTOG Analysis. J Clin Oncol 2008, 26, 3582–3589, doi:10.1200/jco.2007.14.8841.
  2. Hanasoge, S.; Magliocca, K.R.; Switchenko, J.M.; Saba, N.F.; Wadsworth, J.T.; El‐Deiry, M.W.; Shin, D.M.; Khuri, F.; Beitler, J.J.; Higgins, K.A. Clinical Outcomes in Elderly Patients with Human Papillomavirus–Positive Squamous Cell Carcinoma of the Oropharynx Treated with Definitive Chemoradiation Therapy. Head Neck 2016, 38, 846–851, doi:10.1002/hed.24073.
